# Network design principle for robust oscillatory behaviors with respect to biological noise

**Lingxia Qiao[1†], Zhi-Bo Zhang[2,3†], Wei Zhao[2†], Ping Wei[2,4]\*, Lei Zhang[1,2]\***

[1]Beijing International Center for Mathematical Research, Peking University, Beijing, China; [2]Center for Quantitative Biology, Peking University, Beijing, China; [3]Peking-Tsinghua Joint Center for Life Sciences, Academy for Advanced Interdisciplinary Studies, Peking University, Beijing, China; [4]Center for Cell and Gene Circuit Design, CAS Key Laboratory of Quantitative Engineering Biology, Shenzhen Institute of Synthetic Biology, Shenzhen Institutes of Advanced Technology, Chinese Academy of Sciences, Shenzhen, China

**Abstract** Oscillatory behaviors, which are ubiquitous in transcriptional regulatory networks, are often subject to inevitable biological noise. Thus, a natural question is how transcriptional regulatory networks can robustly achieve accurate oscillation in the presence of biological noise. Here, we search all two- and three-node transcriptional regulatory network topologies for those robustly capable of accurate oscillation against the parameter variability (extrinsic noise) or stochasticity of chemical reactions (intrinsic noise). We find that, no matter what source of the noise is applied, the topologies containing the repressilator with positive autoregulation show higher robustness of accurate oscillation than those containing the activator-inhibitor oscillator, and additional positive autoregulation enhances the robustness against noise. Nevertheless, the attenuation of different sources of noise is governed by distinct mechanisms: the parameter variability is buffered by the long period, while the stochasticity of chemical reactions is filtered by the high amplitude. Furthermore, we analyze the noise of a synthetic human nuclear factor κB (NF-κB) signaling network by varying three different topologies and verify that the addition of a repressilator to the activator-inhibitor oscillator, which leads to the emergence of high-robustness motif—the repressilator with positive autoregulation—improves the oscillation accuracy in comparison to the topology with only an activator-inhibitor oscillator. These design principles may be applicable to other oscillatory circuits.

**\*For correspondence:**
ping.wei@siat.ac.cn (PW);
zhangl@math.pku.edu.cn (LZ)

[†]These authors contributed equally to this work

**Competing interest:** The authors declare that no competing interests exist.

## Editor's evaluation

The authors study the important problem of how to achieve accurate oscillation robustly in biological networks where noise level may be high. The authors adopted a comprehensive approach and study how different network configurations affect oscillation. This work makes an important contribution to the field as it offers the first comprehensive survey of networks motifs capable of oscillation, with further characterization of their robustness.

## Introduction

Oscillatory behaviors have been observed in a broad range of biological processes, such as cell cycle (*Ferrell et al., 2011*; *Tyson, 1991*), circadian rhythms (*Partch et al., 2014*), and mitotic wave in *Drosophila* embryo (*Deneke et al., 2016*). Oscillatory features, including period and amplitude, can encode functional information, which plays an essential role in coordinating gene regulation (*Cai

*et al., 2008*) or transmitting distinct stimuli (*Hao and O'Shea, 2012*; *Heltberg et al., 2019*). In past decades, negative feedback, time delay, and nonlinearity have been identified as key mechanisms for biochemical oscillation (*Novák and Tyson, 2008*), following which researchers artificially synthesized biochemical networks capable of oscillation (*Atkinson et al., 2003*; *Chen et al., 2015*; *Elowitz and Leibler, 2000*; *Potvin-Trottier et al., 2016*; *Stricker et al., 2008*; *Tigges et al., 2010*; *Zhang et al., 2017*). Repressilator (*Elowitz and Leibler, 2000*) and activator-inhibitor oscillator (*Atkinson et al., 2003*) are the most famous of these synthetic oscillators.

While many synthetic biological circuits can oscillate, their dynamics are typically irregular, owing to ubiquitous biological noise such as fluctuations in the microenvironment and inherent stochasticity of chemical reactions (*Elowitz et al., 2002*; *Li et al., 2009*; *Potvin-Trottier et al., 2016*; *Raser and O'Shea, 2004*; *Swain et al., 2002*; *Yu et al., 2018*). Thus, a natural question is how the biological systems achieve accurate oscillation in the presence of noise. Previous studies revealed that many kinetic parameters can influence the robustness of the biological oscillators, such as the system size and degree of cooperativity of reactions (*Gonze et al., 2002a*), timescale of the promoter interaction (*Forger and Peskin, 2005*), repressor degradation rate (*Potvin-Trottier et al., 2016*), free energy cost measured by ATP/ADP ratios (*Cao et al., 2015*; *Fei et al., 2018*; *Qin et al., 2021*), and kinetic parameter-determined oscillation mechanisms (i.e., limit cycle or force driving) (*Monti et al., 2018*). Moreover, growing evidence suggests the existence of the relationship between network configurations and noise buffering capabilities for biochemical oscillators. For example, in a synthetic microbial consortium oscillator composed of two different types of bacteria, adding negative autoregulation to the negative feedback loop increases the parameter space to oscillate persistently in the face of noise (*Chen et al., 2015*); an additional positive feedback loop in the biochemical oscillator consisting of the negative feedback loop can decrease the coefficient of variation (CV) of period when considering the stochasticity of reactions (*Mather et al., 2009*) and possess nearly constant period when varying the synthesis rate (*Stricker et al., 2008*).

Instead of exploring mechanisms to achieve accurate oscillation case by case, we try to understand the general network design principles of accurate oscillation using the bottom-up approach (*Ma et al., 2009*; *Qiao et al., 2019*) and discover the specific network topologies that can oscillate and attenuate noise simultaneously. Here, we systematically explore the relationship between the network topology and robustness to different sources of noise in both two- and three-node networks. We first perform an exhausting search of two- and three-node network topologies to identify those capable of oscillation in the absence of noise, and then investigate the abilities of those oscillatory topologies to achieve accurate oscillation in the presence of different sources of noise. Two different sources are considered: parameters are perturbed by noise terms whose magnitudes are proportional to parameters (i.e., extrinsic noise); chemical reactions induce stochasticity due to a small copy number of proteins (i.e., intrinsic noise). We classify all oscillatory topologies according to what core motifs they include, and then compare the ability to execute accurate oscillation in the presence of noise among different categories. Two categories whose core motifs include a repressilator with a positive feedback perform better than others. Importantly, the existence of positive autoregulation always enhances the performance. While these results hold regardless of what source of noise exists, mechanisms to attenuate different sources of noise are distinct: long period buffers the extrinsic noise, and high amplitude attenuates the intrinsic noise. Moreover, we experimentally validate that adding a repressilator to the activator-inhibitor topology in synthetic NF-κB signaling circuits can improve the performance to buffer noise, indicating the important role of the repressilator with a positive autoregulation in filtering noise.

## Results

### Searching for topologies robustly executing accurate oscillation

#### Index for measuring the oscillation accuracy

To measure the accuracy of the oscillatory behavior, we use the dimensionless correlation time, which is the correlation time $\tau$ divided by the period $T$. The correlation time $\tau$ describes how fast the autocorrelation function $C(t)$ exponentially decays. To be specific, for a noisy dynamic trajectory of the oscillator, $C(t)$ displays a damped oscillation (*Figure 1A*):

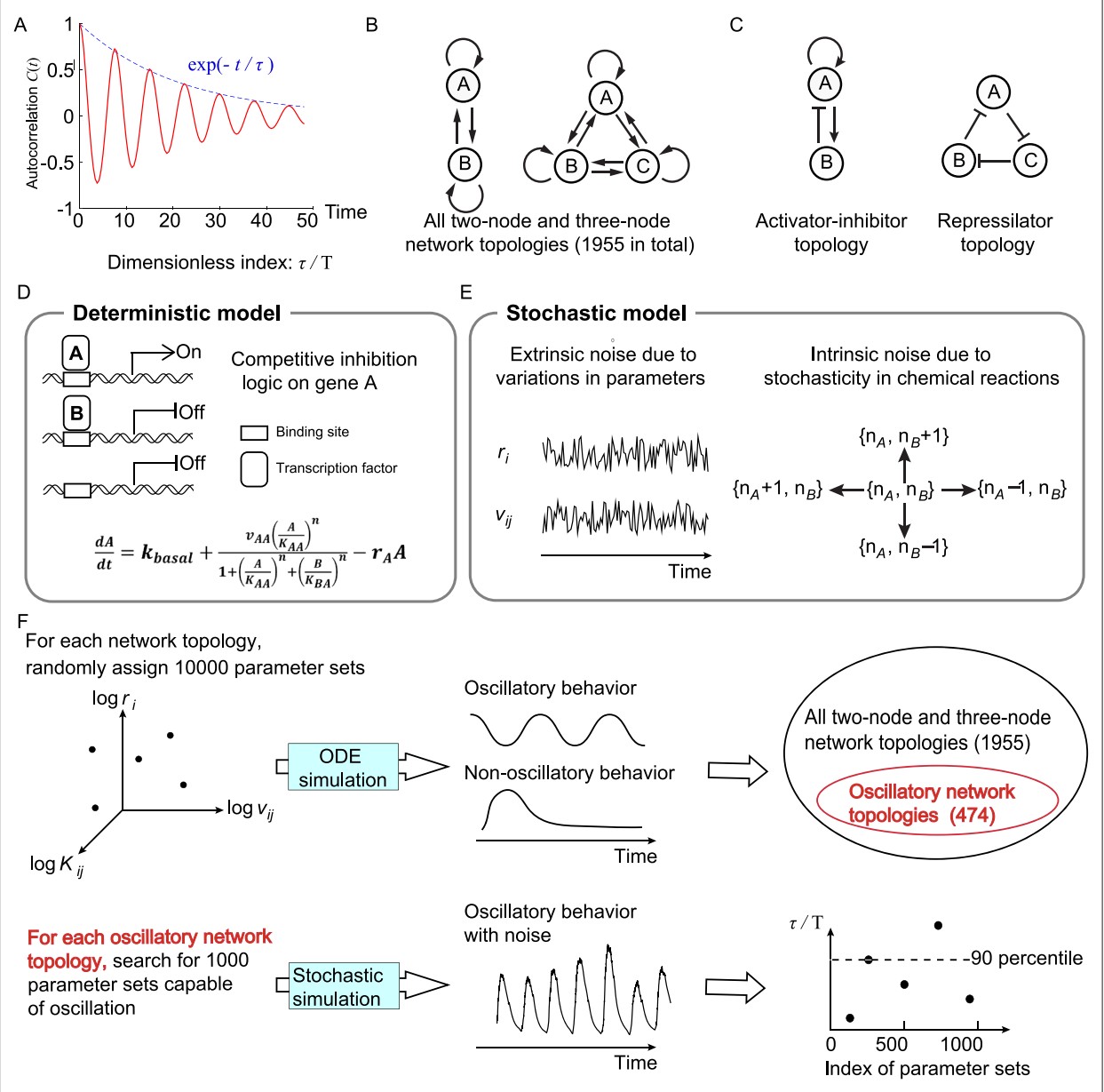

**Figure 1.** Searching all possible two-node and three-node network topologies for oscillatory topologies with high accuracy in the presence of noise. (**A**) The accuracy of the oscillatory behavior against the noise is measured by the ratio of the correlation time $\tau$ to the period $T$. (**B**) Possible links in two-node and three-node network topologies. (**C**) Two typical topologies of biological oscillators: the activator-inhibitor and the repressilator topologies. (**D**) The deterministic model. (**E**) Stochastic models where the extrinsic and intrinsic noise are considered separately. (**F**) Illustration of searching oscillatory topologies and measuring the robustness to noise for a given oscillatory topology. See 'Methods' and *Supplementary file 1a* for parameter ranges and magnitudes of extrinsic and intrinsic noise.

$$C\left(t\right) \equiv \frac{\left\langle\left(x\left(t+s\right)-\langle x\rangle\right)\left(x\left(s\right)-\langle x\rangle\right)\right\rangle_s}{\langle x^2\rangle - \langle x\rangle^2} = \exp\left(-t/\tau\right) \times \cos\left(2\pi t/T\right)$$

where $\langle\cdots\rangle_s$ is defined by $\langle f(s)\rangle_s = \lim\limits_{S\to\infty} \frac{1}{S}\int_0^S f(s)\,ds$, and $\langle\cdots\rangle$ is the ensemble average; $T$ is the period (time needed from one peak to the next peak). If fluctuations of the noisy trajectory are small, the autocorrelation decays slowly, leading to a large value of $\tau$. The correlation time $\tau$ has the same unit as that of the period, so $\tau/T$ is dimensionless. Therefore, instead of using $\tau$, we utilize $\tau/T$ to measure the accuracy, which is equal to the quantity that previous work has used (*Cao et al., 2015*) except a constant factor.

## Network topology space

We limit ourselves to network topologies with two or three nodes (*Figure 1B*) and search for topologies capable of accurate oscillation using a bottom-up concept. While the signaling pathway of the oscillator in nature is complex, the core motif executing functions may be simple (*Lim et al., 2013*; *Ma et al., 2009*; *Novák and Tyson, 2008*; *Qiao et al., 2019*), and thus two- or three-node networks might be enough to capture key features. Besides, the number of all two- and three-node network topologies is $3^9$ because there are nine links in total and each link has three options: activation, inhibition, or does not exist; however, by excluding topologies with isolated nodes or symmetrical to existing topologies, the number of possible two- and three-node network topologies is reduced from $3^9$ to 1955. Here, two typical oscillatory topologies are shown in *Figure 1C*: the activator-inhibitor and repressilator topologies. For the activator-inhibitor topology, the activator (node A) has a positive autoregulation and positively regulates the inhibitor (node B), but is negatively regulated by the inhibitor; for the repressilator topology, each node acts as a repressor to inhibit its next node, thus constituting a cyclic negative-feedback loop.

## Mathematical modeling

To model two- and three-node network topologies, we use transcriptional regulations to describe interactions among nodes (*Figure 1D*; see 'Methods'). In a transcriptional regulatory network, nodes and links represent genes' products and transcriptional regulations, respectively; genes' products work as transcription factors to interact with the regulatory sequence of other genes and activate or inhibit the transcription, regulating the production rates of other genes' products, that is, other nodes. Moreover, when multiple transcription factors regulate the same gene simultaneously, the competitive inhibition logic is adopted: those transcription factors compete for the same binding sites. Thus, the transcriptional activity of a gene depends on the relative weights of transcription factors activating this gene and those inhibiting this gene. *Figure 1D* illustrates the ordinary differential equation describing dynamics of node A when node A not only activates itself but also is inhibited by node B. In this equation, the variable $A$ represents the concentration of the product of gene A; $k_{basal}$ is the basal production rate (much smaller than other terms); $v_{AA}$ is the maximum production rate caused by product A; $K_{AA}$ and $K_{BA}$ are binding affinities of products $A$ and $B$ to gene A, respectively; $r_A$ is the degradation rate; $n$ is the Hill coefficient; and the production rate is determined by relative weights of $A$ and $B$.

Based on the above deterministic model, we develop stochastic models to describe the oscillatory behavior in the presence of noise. According to the source of noise, the biological noise can be decomposed into extrinsic and intrinsic components. On the one hand, we model the extrinsic noise as the variability of parameters including the maximum production rate $v$ and the degradation rate $r$ (*Figure 1E*; see 'Methods'): each of these parameters is added by a noise term with zero mean, and the standard deviation of the noise term is proportional to the value of the kinetic parameter. On the other hand, the intrinsic noise, generated by the stochasticity of discrete chemical reactions, is modeled by directly simulating the dynamics of molecular numbers rather than concentrations. To this end, we introduce the cell volume $V$, and naturally the molecular number of each node is the product of the cell volume $V$ and the concentration. As reactions progress, the molecular numbers would randomly increase or decrease by one at some time point (*Figure 1E*; see 'Methods'), and the waiting time of the increase and decrease obeys exponential distributions with parameters determined by the production and decay rates in the deterministic model, respectively. This stochastic process can be exactly solved by the Gillespie algorithm, which has been widely used in previous studies (*Liu et al., 2020*; *Thattai and van Oudenaarden, 2001*; *Veliz-Cuba et al., 2015*; *Zhao et al., 2021*); however, the computation cost is high, and thus we use chemical Langevin equations as approximations to reduce the cost (*Gillespie, 2000*). Although the biological noise in nature usually has the extrinsic and intrinsic components simultaneously, we only consider the case where only one source of noise exists for simplicity, that is, only extrinsic noise exists or only intrinsic noise exists.

## Procedures to search for network topologies robustly executing accurate oscillation

To search for two- and three-node network topologies that can robustly achieve accurate oscillation (i.e., high dimensionless correlation time $\tau/T$), two steps are performed (*Figure 1F*): the first step is

to identify topologies capable of oscillation in the whole network topology space (the upper panel in *Figure 1F*); the second step is to use the 90-percentile value of $\tau/T$ to quantify the robustness of each oscillatory network topology to achieve accurate oscillation (the lower panel in *Figure 1F*). For a given topology, the 90-percentile value of $\tau/T$ is defined as the value of $\tau/T$ below which 90% of $\tau/T$'s fall when 1000 parameter sets are randomly assigned. We refer the reader to 'Methods' for details, and here we only show major procedures. In the first step (the upper panel in *Figure 1F*), to obtain oscillatory network topologies in the whole network topology space, we randomly assign 10,000 parameter sets for each network topology and simulate the deterministic dynamics. The oscillatory network topology is chosen by the following two criteria: the network topology without repressilator is regarded as an oscillatory network topology if at least 80 parameter sets are capable of oscillation; the network topology with repressilator is defined as an oscillatory network topology if at least 10 parameter sets achieve oscillation. In this way, we finally obtain 474 oscillatory network topologies, and nearly 35% of them are with the repressilator. If we used the threshold of 80 for all network topologies, oscillatory network topologies with repressilator only occupy 20% of all oscillatory network topologies, which may lose the generality of conclusions about the repressilator. In the second step (the lower panel in *Figure 1F*), for each of these 474 oscillatory network topologies, we sample 1000 parameter sets capable of oscillation in the absence of noise and calculate the 90-percentile value of $\tau/T$ in the presence of extrinsic noise or intrinsic noise. This value measures the robustness of the given topology against noise: the higher the value is, the larger probability to achieve accurate oscillation the topology has.

## The robustness of accurate oscillation against extrinsic noise for different network topologies

### Classification of all 474 oscillatory network topologies

We start by classifying all 474 oscillatory network topologies according to five types of core motifs. These five types of core motifs are as follows: the first core motif (shown in brown in *Figure 2A*) is composed of the repressilator and a positive autoregulation, but the node with the positive autoregulation is not allowed to have a positive incoming link; the second core motif (shown in orange in *Figure 2A*) is similar to the first core motif except that the positive incoming link to the positive autoregulated node is required; the third type of core motifs include the activator-inhibitor topology and its two variants (shown in green in *Figure 2A*); the fourth and fifth core motifs are the repressilator and delayed negative feedback (*Figure 2A*), respectively. Based on the identification of five types of core motifs, we define C1 category as the network topologies that contain only the first type of core motif, and so do the C2 category, C3 category, C4 category, and C5 category. The above five categories constitute near 59% of all 474 oscillatory network topologies, while the rest of topologies are those containing at least two of these five types of core motifs. Note that these oscillatory network topologies all have a negative feedback structure, which is consistent with previous studies (*Glass and Pasternack, 1978*; *Novák and Tyson, 2008*).

## The topologies containing repressilator with positive autoregulation perform better than those containing the activator-inhibitor topology when facing extrinsic noise, and the positive autoregulation enhances the robustness against extrinsic noise

Next we compare the robustness of accurate oscillation against extrinsic noise among above C1, C2, …, and C5 categories. *Figure 2B* shows the violin plots of 90 percentiles of $\tau/T$ in the presence of extrinsic noise for the five categories, where each violin corresponds to one category. By applying one-tailed Wilcoxon rank-sum tests to adjacent two categories, we find that 90 percentiles of $\tau/T$ for C1 category are significantly larger than those for C2 category, and this relation also holds between C2 and C3 categories, between C3 and C4 categories, and between C4 and C5 categories. These findings indicate that the order of these five categories according to the robustness of accurate oscillation to extrinsic noise is C1 > C2 > C3 > C4 > C5. The facts that C1 > C3 and C2 > C3 demonstrate that topologies containing the repressilator with positive autoregulation achieve higher robustness against extrinsic noise than those containing the activator-inhibitor topology. Besides, core motifs in both C1 and C2 categories have an extra positive autoregulation in comparison to the core motif (the

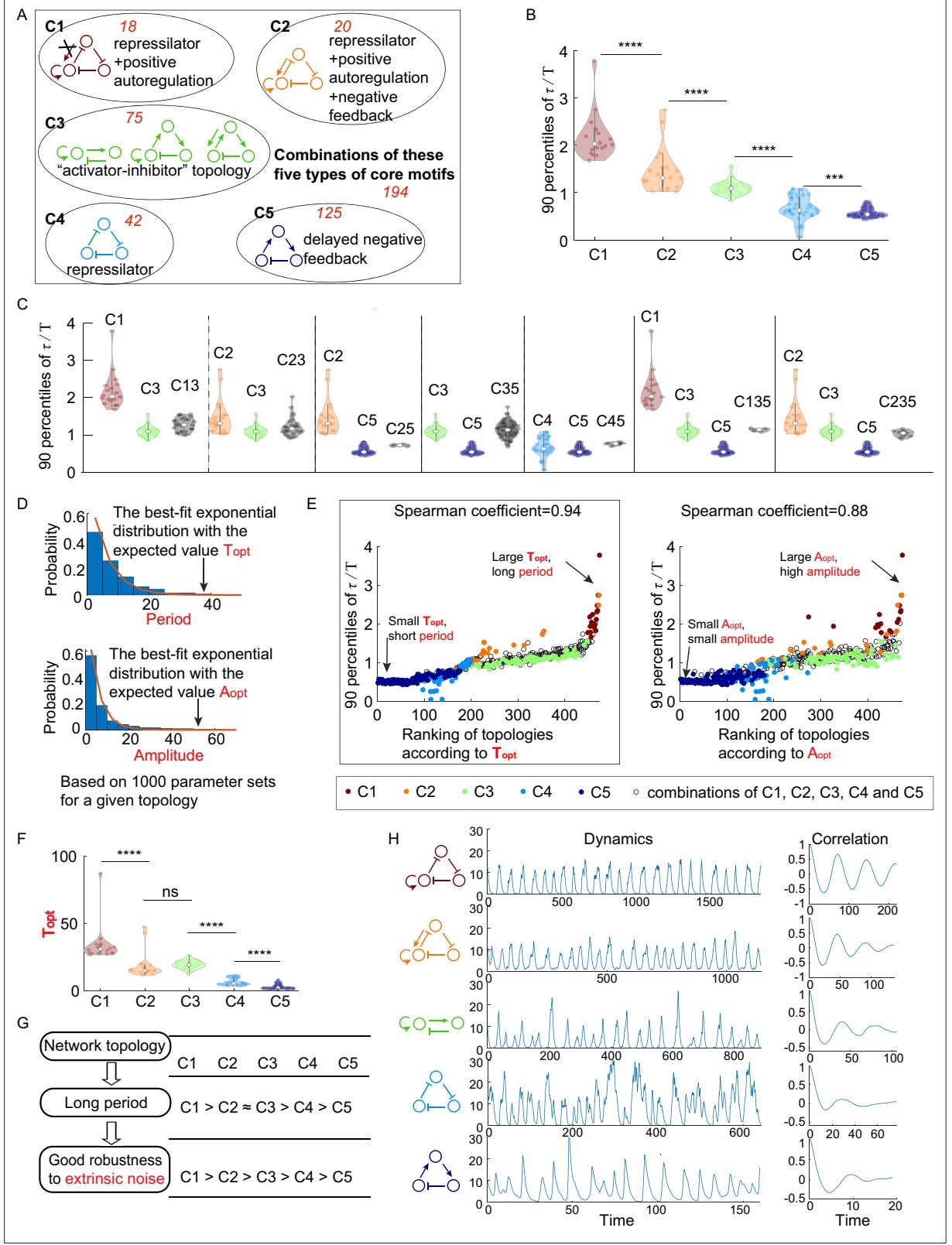

**Figure 2.** The relationship between network topology and robustness to extrinsic noise. (**A**) Venn diagram of all 474 oscillatory network topologies. The C1, C2, C3, C4, and C5 categories are nonoverlapping collections of network topologies that contain core motifs in brown, orange, green, blue, and dark blue, respectively. The number in red is the number of topologies in each region. (**B**) The violin plots of 90 percentiles of dimensionless correlation time ($\tau/T$) for C1, C2, …, and C5 categories present in (**A**). Each category corresponds to one violin plot, and topologies are denoted by dots. The

*Figure 2 continued on next page*

*Figure 2 continued*

Wilcoxon rank-sum tests (one-tailed) are applied to adjacent categories (\*\*\*p<0.001; \*\*\*\*p<0.0001). (**C**) The violin plots of 90 percentiles of $\tau/T$ for topologies that do not belong to any of C1, C2, …, and C5 categories. Taken C13 category as an example, the C13 category is the collection of the topology containing the first and third types of motifs simultaneously. (**D**) Illustration of calculations of the period average (denoted as $T_{opt}$) and the amplitude average (denoted as $A_{opt}$) for a given topology. For a given topology, $T_{opt}$ (or $A_{opt}$) is the expectation of the best-fit exponential distribution of 1000 periods (or 1000 amplitudes) with random assigned parameters. (**E**) The scatter plots of 90 percentiles of $\tau/T$ versus the ranks according to $T_{opt}$ (left) and $A_{opt}$ (right). Each oscillatory network topology is denoted by a dot, with color determined by its category. Spearman coefficients are illustrated on the top of each plot, and the plot with a relatively high Spearman coefficient is highlighted by the black box. (**F**) The violin plots of $T_{opt}$ for the five network topology categories present in (**A**). The Wilcoxon rank-sum tests (one-tailed) are applied to adjacent categories (ns, not significant; \*\*\*p<0.001; \*\*\*\*p<0.0001). (**G**) Hypothesis of topology–period–robustness relation. The network topology may affect robustness to extrinsic noise by acting on an intermediate quantity – period. (**H**) Typical dynamics in the presence of extrinsic noise and corresponding autocorrelation functions for the five topologies present in (**A**). From top to bottom, the period length decreases, and the autocorrelation also decreases. The amplitude for those dynamics in the absence of noise is almost the same (near 12). The 90 percentile of $\tau/T$ is the average from five replicates.

The online version of this article includes the following figure supplement(s) for figure 2:

**Figure supplement 1.** The same scatter plots as *Figure 2E*, except that each subplot represents one category.

**Figure supplement 2.** The distributions of $\tau/T$ for 20 oscillatory network topologies when randomly sampling 1000 kinetic parameter sets.

repressilator) in C4 category, suggesting that the higher robustness of C1 and C2 categories than C4 category against extrinsic noise is due to the effect of positive autoregulation in improving the robustness to extrinsic noise. This effect is also validated by the comparison of the robustness between C3 and C5 categories.

The above analyses focused on the oscillatory network topologies in C1, C2, …, and C5 categories, which account for nearly 59% of all oscillatory network topologies. To perform a complete research, we also investigate the robustness to extrinsic noise for the remaining 41% oscillatory network topologies. These topologies all contain at least two types of core motifs and can be classified into seven categories, based on what core motifs the topology has. Then we compare each of them with its 'component' category (i.e., C1, C2, …, or C5 categories). For example, C13 category is composed of topologies that contain both the first and third types of core motifs, and its two 'component' categories are defined as C1 and C3 categories. The comparison is made in *Figure 2C*, where each group of violin plots separated by dashed lines represents the 90 percentiles of $\tau/T$ (in the presence of extrinsic noise) for the category with combined core motifs and its 'component' categories. It can be seen that the category with combined core motifs usually shows intermediate robustness among its 'component' categories. That is to say, if a network topology has low robustness against extrinsic noise, adding a high-robustness core motif usually improves the robustness, but the combined topology cannot outperform the added high-robustness core motif.

## Topologies with long period achieve high robustness against extrinsic noise

The above analyses indicate that network topologies differ widely in their robustness to achieve accurate oscillation, then we ask what mechanisms cause these differences. Note that how the system responds to the noise is often linked to the deterministic features (*Monti et al., 2018*; *Paulsson, 2004*; *Wang et al., 2010*). For example, Monti et al. found that the circuit's ability to sense time under input noise becomes worse when this circuit's deterministic behavior cannot generate the limit cycle; Wang et al. adopted a similar form of noise and demonstrated the importance of signed activation time, a quantity calculated based on deterministic behavior, on the noise attenuation; by using an $\Omega$-expansion to approximate the birth-and-death Markov process, Paulsson obtained the variance of the protein in gene networks and found that it is related to the network's elasticity, which is calculated from the deterministic model. Based on these observations, we explore two important characteristics for the oscillator: period and amplitude. Instead of focusing on a specific oscillatory network topology, we consider all 474 oscillatory network topologies and study what period and amplitude each topology prefers. To be precise, for each network topology, we calculate the distributions of period and amplitude from 1000 randomly sampled oscillation parameter sets and approximate mean values of period and amplitude by $T_{opt}$ and $A_{opt}$, respectively. Here, we refer to the amplitude as the maximal peak value among nodes A–C; $T_{opt}$ (or $A_{opt}$) is defined as the expectation of the best-fit exponential distribution of 1000 periods (or 1000 amplitudes) (*Figure 2D*). Therefore, the topology with large $T_{opt}$ tends to oscillate with long period, and the topology with large $A_{opt}$ usually indicates an

oscillation with high amplitude. Note that these two quantities are calculated in the noise-free system, and thus are not affected by the amplitude of the noise source or the type of noise.

To investigate the role of above two quantities $T_{opt}$ and $A_{opt}$ in the robustness of accurate oscillation against extrinsic noise, we calculate Spearman coefficients between these two quantities and 90 percentiles of $\tau/T$ for all 474 oscillatory network topologies (*Figure 2E*). In *Figure 2E*, each dot represents an oscillatory network topology, with the x-axis representing the ranking according to $T_{opt}$ (left panel) or $A_{opt}$ (right panel). The Spearman coefficient between $T_{opt}$ and 90 percentile of $\tau/T$ for all 474 oscillatory network topologies is 0.94, which is larger than that between $A_{opt}$ and 90 percentile of $\tau/T$ (0.88). This result not only holds for all 474 oscillatory network topologies, but also holds within each of C1, C2, …, and C5 categories (*Figure 2—figure supplement 1*). These findings indicate that the robustness to extrinsic noise is more highly correlated with long period rather than high amplitude.

Since the long period benefits the robustness to extrinsic noise, then we ask how network topologies affect the period and whether those topologies with long period indeed show high robustness to extrinsic noise. To answer these questions, we analyze $T_{opt}$ for C1, C2, …, and C5 categories (*Figure 2F*). The ranking of these five categories according to $T_{opt}$ is C1 > C2 ≈ C3 > C4 > C5, which is obtained by the one-tailed Wilcoxon rank-sum tests for each adjacent two categories. This ranking is almost the same as that according to the robustness of accurate oscillation to extrinsic noise (C1 > C2 > C3 > C4 > C5) except rankings for C2 and C3 categories, suggesting that the topology with long period usually leads to high robustness of accurate oscillation to extrinsic noise (*Figure 2G*). The only inconsistency is that C2 and C3 categories differ in the robustness but have no significant difference in the probability to achieve long period. That is to say, C2 category might show better robustness to extrinsic noise than C3 category though they have the same period. *Figure 2H* shows typical dynamics for five different topologies when extrinsic noise exists. Those topologies from the top panel to the bottom panel belong to categories C1 to C5, respectively. Their dynamics have almost the same amplitude, but the period, as well as the autocorrelation, decreases when categories vary from C1 to C5. These findings suggest that topologies with prolonged period tend to have good performance to filter extrinsic noise, and this correlation is less likely due to that they have different amplitude.

## The robustness of accurate oscillation against intrinsic noise for different network topologies

### In the presence of only intrinsic noise, the repressilator with positive autoregulation is still better than the activator-inhibitor, and the advantage of positive autoregulation still holds

Unlike considering the robustness of accurate oscillation against parameter variability in the previous section, we next study the case where only intrinsic noise exists. With the same oscillatory network topology categories present in *Figure 2A*, 90 percentiles of the dimensionless correlation time ($\tau/T$) in the presence of only intrinsic noise also show a roughly similar trend from C1 to C5 categories (*Figure 3A*) except that C2 category exhibits the same robustness with C1 category to intrinsic noise while C2 category shows lower robustness than C1 in the presence of extrinsic noise. Moreover, the higher robustness of C1 and C2 categories compared with C3 category validates the better performance of the repressilator with positive autoregulation than the activator-inhibitor topology; the improvement of robustness from C4 category to C1 (or C2) category indicates the effect of positive autoregulation on the robustness to intrinsic noise; the comparison between the robustness of C5 and C3 categories also implies the advantage of the positive autoregulation. These findings are consistent with those when noise is only originated from the extrinsic noise. Another consistency is that the hybrid of core motifs imparts an intermediate robustness not only in the presence of the extrinsic noise (*Figure 2C*) but also in the presence of the intrinsic noise (*Figure 3B*).

### Topologies with the high amplitude enable high robustness against intrinsic noise

Similar to the analysis of robustness to extrinsic noise, we try to answer whether the period or amplitude is highly correlated with the robustness to intrinsic noise. In the presence of only intrinsic noise, the Spearman correlation coefficient of 90 percentiles of $\tau/T$ and $T_{opt}$ for all 474 oscillatory network topologies is 0.72, which is smaller than that of 90 percentiles of $\tau/T$ and $A_{opt}$ (0.81) (*Figure 3C*).

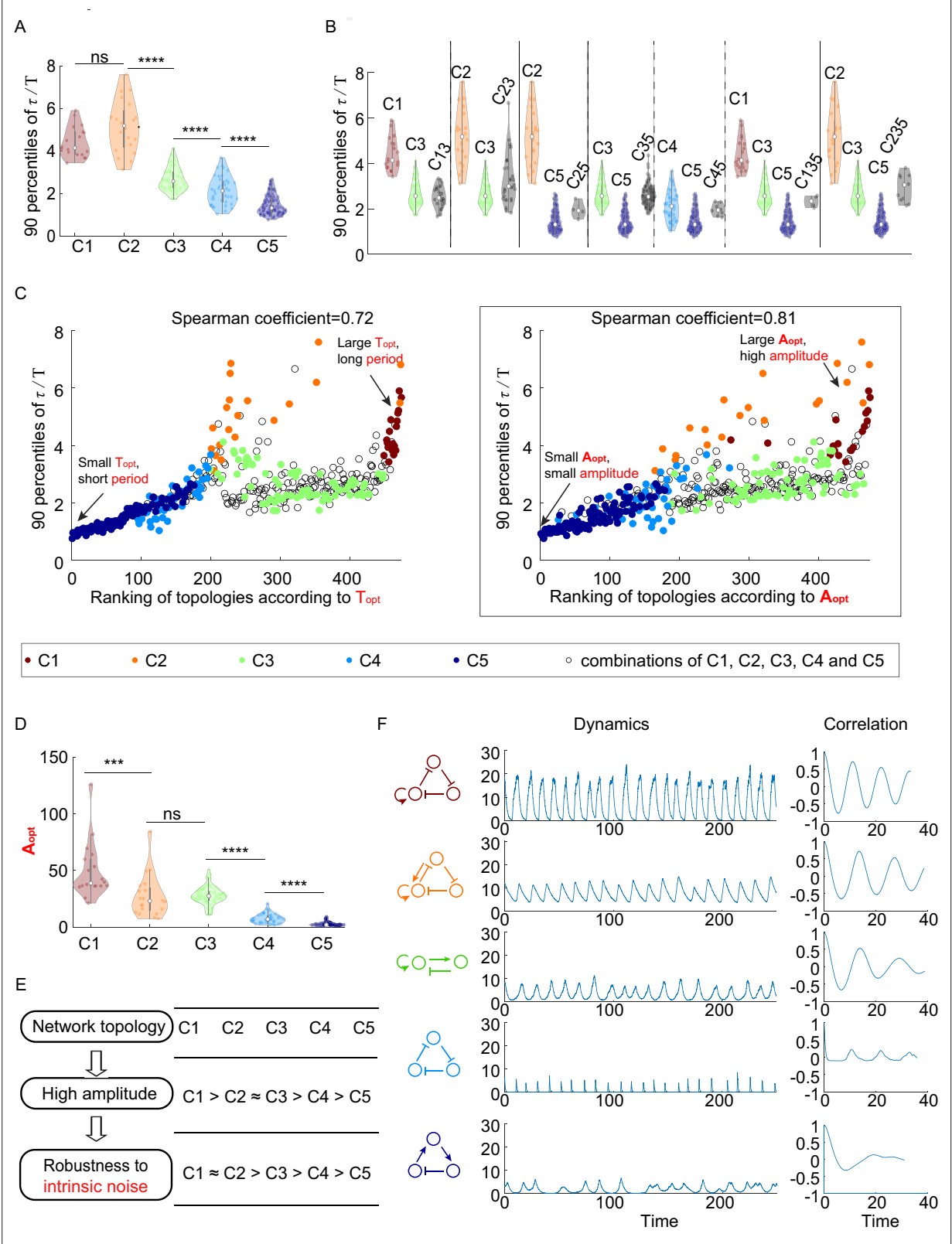

**Figure 3.** The relationship between network topology and robustness to intrinsic noise. (**A–C**) The same plots as in *Figure 2B, C and E*, except that the noise comes from the stochasticity of biochemical reactions. The 90 percentile of dimensionless correlation time ($\tau/T$) shows a higher correlation with $A_{opt}$ rather than $T_{opt}$. (**D**) The violin plots of $A_{opt}$ for the five network topology categories present in *Figure 2A*. The Wilcoxon rank-sum tests (one-tailed) are applied to adjacent categories (ns, not significant; ***p<0.001; ****p<0.0001). (**E**) Hypothesis of topology–amplitude–robustness relation. The

*Figure 3 continued on next page*

*Figure 3 continued*

correlation between $A_{opt}$ and 90 percentiles of $\tau/T$ in the presence of intrinsic noise indicates that the network topology may influence the robustness against intrinsic noise through the amplitude. (**F**) The same plot as *Figure 2H* except in the presence of intrinsic noise. From top to bottom, while the period in the absence of noise is very close (near 12), the amplitude decreases, and the autocorrelation also decreases. The 90 percentile of $\tau/T$ is the average from five replicates.

The online version of this article includes the following figure supplement(s) for figure 3:

**Figure supplement 1.** The same plots as *Figure 3C*.

**Figure supplement 2.** The relationship between network topology and robustness to intrinsic noise when using the Gillespie algorithm to simulate the dynamics.

**Figure supplement 3.** Comparisons of dimensionless correlation time obtained from different simulation methods.

**Figure supplement 4.** Same plot as *Figure 2—figure supplement 2* except that the intrinsic noise is considered.

These findings suggest that unlike the case of extrinsic noise where the robustness is more strongly correlated with period, the robustness of accurate oscillation against intrinsic noise is more highly correlated with amplitude. In other words, the topology with the high amplitude has a larger probability to achieve high robustness against intrinsic noise than that with long period. However, it should be noted that the correlation coefficient between 90 percentiles of $\tau/T$ and $A_{opt}$ for all oscillatory network topologies is not very close to 1 (the right panel in *Figure 3C*), and it is also much smaller than 1 even in each category (*Figure 3—figure supplement 1*), implying that the relation between the amplitude and robustness to intrinsic noise is not very strong, and that some topologies with small amplitude may perform better than those with high amplitude. Therefore, there might exist other mechanisms to attenuate intrinsic noise.

Furthermore, by applying Wilcoxon rank-sum tests to amplitude average ($A_{opt}$) for neighboring two categories, we find that the ranking of five network topology categories according to amplitude is C1 > C2≈ C3 > C4 > C5 (*Figure 3D*). This ranking is almost the same as that according to the robustness to intrinsic noise (C1 ≈ C2 > C3 > C4 > C5) (*Figure 3E*), implying that the amplitude might link the topology category and the robustness to intrinsic noise. The only exception is C2 category: because of the fact that C1 > C2 ≈ C3 according to $A_{opt}$ and the fact that the amplitude strongly correlates with the robustness, the C2 category is supposed to show the same robustness to intrinsic noise as C3 category and exhibit lower robustness than C1 category; however, the robustness to intrinsic noise for C2 category is actually at the same level of C1 category, further demonstrating that the high amplitude is not the only mechanism to enhance the robustness to intrinsic noise (*Figure 3—figure supplement 1*). *Figure 3F* shows typical dynamics in the presence of intrinsic noise whose topologies belong to distinct categories. Those dynamics exhibit near period, but their amplitudes and autocorrelations decrease from category C1 to category C5, which supplies a possibility to enhance the robustness to intrinsic noise through varying topologies while maintaining period.

## Simulations using the Gillespie algorithm lead to similar conclusions

The above analyses are based on simulations for chemical Langevin equations, which can only give approximate solutions of the dynamical behavior in the presence of intrinsic noise. To test whether this approximation is feasible, we use the Gillespie algorithm to exactly solve the stochastic dynamical behavior when facing intrinsic noise, and then conduct similar analyses (*Figure 3—figure supplement 2*) as the previous section has done. According to the robustness rankings for C1–C5 categories, the repressilator with positive autoregulation performs better than the activator-inhibitor, and the topologies with positive autoregulation are better than that without positive autoregulation. Besides, the robustness is more correlated with the mean amplitude rather than the mean period, and the order of the five categories sorted by the mean amplitude is almost the same as that sorted by robustness, indicating the bridge role of amplitude to link topologies and the robustness to intrinsic noise. These results are consistent with the conclusions based on chemical Langevin equations. We also find that the Gillespie algorithm leads to higher dimensionless correlation times than chemical Langevin equations since the maximal correlation in *Figure 3A* is near 6 and that in *Figure 3—figure supplement 2A* is 40. However, this difference does not indicate that chemical Langevin equations are bad approximations: when the system behaves normal noise filtering capability, these two methods give similar dimensionless correlation times (*Figure 3—figure supplement 3A*); when the system buffer

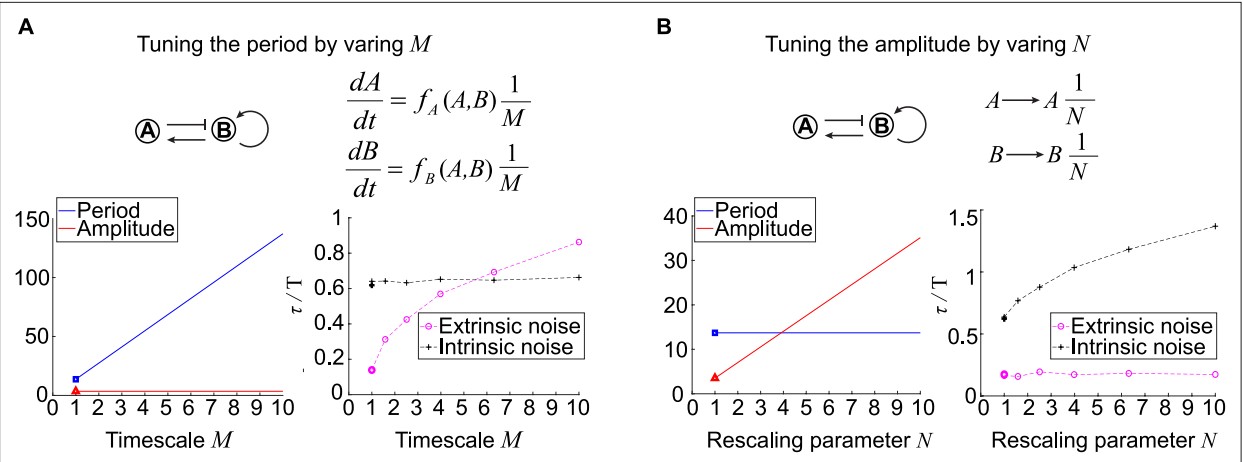

**Figure 4.** Effects of tuning period or amplitude on oscillation accuracies. (**A, B**) Relations between period (**A**) or amplitude (**B**) and oscillation accuracies. For the activator-inhibitor topology, we arbitrarily choose an oscillation parameter set (see *Supplementary file 1b* for parameters), and then we add $M$ or $N$ in the way shown in the upper panel to tune the period or amplitude, respectively. The lower panels are simulation results of the period, amplitude, and dimensionless correlation time when $M$ or $N$ increases.

The online version of this article includes the following figure supplement(s) for figure 4:

**Figure supplement 1.** The mean period and mean amplitude for all 474 oscillatory network topologies show a strong positive correlation.

noise perfectly, dimensionless correlation times calculated through these two methods differ a lot, but autocorrelation functions remain similar (*Figure 3—figure supplement 3B*), which indicates that chemical Langevin equations still capture the system's ability to buffer noise. The reason why large and extremely large dimensionless correlation times result in almost same correlations might be that doubling long correlation time cannot increase autocorrelation efficiently due to the property of the exponential function.

## Relations between period/amplitude and oscillation accuracy against noise are validated by analytical approaches

The above simulations revealed relations between two important features (i.e., period and amplitude) of the oscillator and the oscillator's robustness to noise. However, these results just showed the correlation rather than the causal relationship. Besides, because the period and amplitude are usually positively correlated (*Figure 4—figure supplement 1*), it is hard to control one feature and analyze the effect of the other feature. Fortunately, these two problems can be solved by introducing the timescale or rescaling parameters. In this way, we can change one feature while maintaining the other feature, and then analytically derive causal relations between period or amplitude and the oscillation accuracy. We will illustrate these methods and corresponding results below.

To study the relation between period and oscillation accuracy against noise, we maintain the amplitude and tune the period through changing the factor $M$ on the right-hand side in ordinary differential equations (*Figure 4A*), and then analyze the phase noise through the analytical approach proposed by *Demir et al., 2000*. Varying $M$ can be regarded as the rescaling of time $t$, so the period is changed while maintaining the amplitude, and thus we can focus on the effect of period on the oscillation accuracy. In order to analyze the system with variable $M$, we first summarize Demir et al.'s work. They carried out nonlinear perturbation analysis for oscillators and obtained an exact equation for phase deviation. We only summarize the main results below. The dynamics of a perturbed oscillator can be described as a set of differential equations:

$$\dot{x} = f(x) + B(x)\xi(t)$$

where $x \in \mathbb{R}^3$, $f(\cdot) : \mathbb{R}^3 \to \mathbb{R}^3$, $B(\cdot) : \mathbb{R}^3 \to \mathbb{R}^{3 \times 3}$ and $\xi(t) \in \mathbb{R}^3$ is random noise. The unperturbed system $\dot{x} = f(x)$ has a periodic solution $x_s(t)$ (with period $T$). It can be proved that the variance of the phase deviation $\sigma^2(t)$ satisfies $\sigma^2(t) = ct$, and $c$ is as follows:

$$c = \tfrac{1}{T} \int_0^T v_1^T\left(t'\right) B\left(x_s\left(t'\right)\right) B^T\left(x_s\left(t'\right)\right) v_1\left(t'\right) dt', \tag{1}$$

where $v_1^T(t)$ is the first row of the matrix $V(t)$. Here, the first column of $V^{-1}(t)$ is $\dot{x}_s(t)$, and $V^{-1}(t)\, diag\left[\mu_1, \mu_2, \mu_3\right] V(t)$ is the state transition matrix for $\dot{w} = A(t) w(t)$ where $\mu_i$'s are Floquet exponents and $A(t) = \frac{\partial f(x)}{\partial x}|_{x=x_s(t)}$ (see 'Methods' for details). The **Equation 01** gives an analytic expression describing the phase noise, so we use the dimensionless $c$, that is, $c/T$, to measure the oscillation accuracy instead of $\tau/T$. To calculate $c/T$ for the systems with $M$, we use $T$ to denote the period for the system without $M$, and then the period for this new system is $MT$. Besides, $v_1(t)$ becomes $Mv_1\left(\frac{t}{M}\right)$ (see 'Methods' for details). For the noise term, we merge $\frac{1}{M}$ with kinetic parameters $v_{ij}, \delta_i, r_i$, that is, these parameters become $\frac{1}{M}$ of original values, and then we model the extrinsic and intrinsic noise as that in **Figure 1E**. Therefore, $B(x)$ becomes $B(x)\frac{1}{M}$ when facing extrinsic noise as the magnitude of noise source is proportional to the kinetic parameters. However, in the presence of only intrinsic noise, $B(x)$ becomes $B(x)\sqrt{\frac{1}{M}}$ because the noise term in the chemical Langevin equation is usually the square root of reaction rates. Then we can calculate the ratio of the slope of the variance of the phase noise to the period ($c/T$) using the **Equation 01** (see 'Methods' for details). We find that the $c/T$ in the presence of only extrinsic noise is proportional to the $1/M$, and that in the presence of only intrinsic noise is not affected by $M$. Note that the smaller the $c/T$ is, the more accurate the oscillation is. Thus, large $M$ enhances the oscillation accuracy against extrinsic noise, which is also numerically validated by the trend of dimensionless correlation times for the system with different $M$ (right lower panel in **Figure 4A**). Since large $M$ leads to long period but has no influence on amplitude, the prolonged period might be the reason for high oscillation accuracy in the presence of extrinsic noise.

For the relation between amplitude and oscillation accuracy against noise, we keep the period and tune the amplitude through the rescaling parameter $N$ and then analyze the rescaled system. For a fixed topology with a set of oscillation kinetic parameters, we replace the variables $A$, $B$, and $C$ with $\widetilde{A}/N$, $\widetilde{B}/N$, and $\widetilde{C}/N$, respectively (**Figure 4B**). This rescaling makes amplitudes of $\widetilde{A}$ $N$ times as high as that of $A$, and so do $\widetilde{B}$ and $\widetilde{C}$. However, this rescaling has no influence on the period, so we can focus on the role of amplitude in the oscillation accuracy. The system with rescaled variables $\widetilde{A}$, $\widetilde{B}$, and $\widetilde{C}$ shows unchanged oscillation accuracy against extrinsic noise with varied $N$, but the oscillation accuracy against intrinsic noise increases with increased $N$ (see 'Methods'). Taken together, large $N$ not only increases the amplitude but also improves the oscillation accuracy to intrinsic noise while maintaining the period. These results are consistent with numerical simulations for tendencies of period, amplitude, and dimensionless correlation times (lower panel in **Figure 4B**). These results indicate that the improvement of the oscillation accuracy to intrinsic noise may due to the high amplitude rather than period.

## Analyses of synthetic NF-κB signaling circuits demonstrate the improvement of the oscillation accuracy when adding a repressilator topology to the activator-inhibitor

In previous sections, we have used two- and three-node networks to approximate biological systems and focused on the noise coming from the variability in kinetic parameters or chemical reactions. Though biological systems are more complex than two- or three-node networks and face noise from various sources besides the above noises, the investigation of a specific biological system— a synthetic NF-κB signaling circuit is consistent with the theoretical results in previous sections. As described in our previous work, we implement the design of negative feedback-only circuit 1 (**Figure 5A**) by integrating the synthetic RelA-IκBα signaling circuit into the yeast MAPK pathway. The nuclear-to-cytoplasmic RelA oscillations can be triggered by inducing the degradation of IκBα through the activation of yeast MAP kinase Fus3, and we can monitor these single-cell oscillations for up to 10 hr. Base on this simple circuit, we then modify its structure by adding extra regulations. One modification is adding constantly expressed IκBα. This copy of IκBα also inhibits RelA and is inhibited by Fus3, so it provides another pathway from Fus3 to activate RelA (the orange link in circuit 2 in **Figure 5A**). Another modification is adding a yeast MAPK phosphatase Msg5 (the orange link in circuit 3 in **Figure 5A**), which is activated by RelA and can dephosphorylate Fus3. In circuit 3, Msg5, RelA, and IκBα form a repressilator topology. To obtain the single-cell time trajectories for these three circuits, we employ time-lapse microscopy to track the RelA nuclear localization dynamics for over 10 hr. The

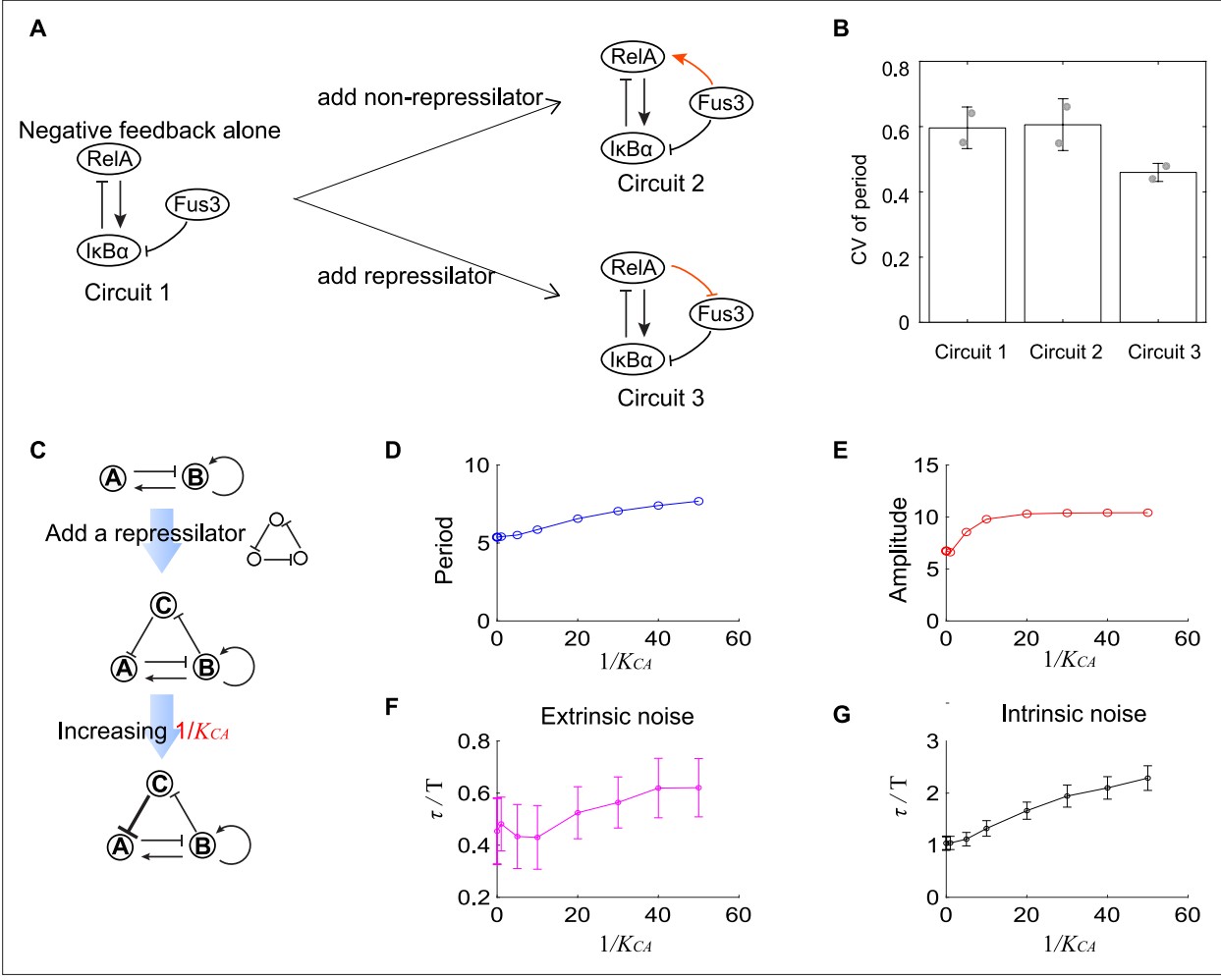

**Figure 5.** Experimental evidence of the improvement of oscillation accuracy when adding a repressilator topology to the activator-inhibitor. (**A**) Schematic showing the synthetic NF-$\kappa$B signaling circuits (circuit 1) and other two circuits (circuits 2 and 3) with more complicated structures. In circuit 1, RelA activates I$\kappa$B$\alpha$, but the latter inhibits RelA, constituting a negative feedback; we regard circuit 1 as the activator-inhibitor where the positive autoregulation is replaced by multiple reactions. Based on circuit 1, we construct circuit 2 by adding a copy of I$\kappa$B$\alpha$ to form a new link from Fus3 to NF-$\kappa$B (RelA), where RelA, I$\kappa$B$\alpha$, and Fus3 cannot constitute a repressilator. Circuit 3 is circuit 1 with additional negative regulation from RelA to Fus3, which is constructed by adding a yeast MAPK phosphatase Msg5 that is activated by RelA and dephosphorylate Fus3. Circuit 3 includes a repressilator consisting of RelA, I$\kappa$B$\alpha$, and Fus3. (**B**) Bar plots of coefficients of variation (CVs) of period for three synthetic circuits present in (**A**). For each circuit, time required from the peak of RelA to next peak from 0 to 10 hr for over 50 cells is recorded to calculate the CV. Two replicates are performed for each circuit. (**C**) Procedures to study how adding repressilator topology to activator-inhibitor affect the oscillation accuracy by using mathematical models present in *Figure 1*. $K_{CA}$ represents the binding affinity of protein C to gene A, and large $1/K_{CA}$ indicates strong inhibition from protein C to protein A. (**D–G**) Simulation results for period (**D**), amplitude (**E**), and dimensionless correlation time in the presence of extrinsic (**F**) or intrinsic (**G**) noise when $1/K_{CA}$ increases. See *Supplementary file 1c* for parameters. The marker with x coordinate 0 denotes the value for activator-inhibitor. The error bar represents the standard deviation for 100 repeated simulations. Increasing $1/K_{CA}$ (i.e., strengthening the inhibition from protein C to protein A) prolongs period, increases the amplitude, and improves the oscillation accuracy.

period lengths are determined as the time intervals between the successive peaks of these trajectories, and then we calculate the CV of those period lengths for over 50 cells. We find that the circuit 2 shows similar CV of period as that for the circuit 1, but that circuit 3 exhibits lower CV of period than circuit 1 (*Figure 5B*). These results suggest that the additional repressilator topology indeed facilitates the noise buffering capability for the activator-inhibitor topology.

Such improvement of the oscillation accuracy when adding a repressilator topology to the activator-inhibitor in the synthetic NF-κB circuit can also be validated using the mathematical models as described in *Figure 1*. We use nodes A, B, and C to denote IκBα, RelA, and Fus3, respectively, and thus circuits 1 and 3 in *Figure 5A* are networks shown in upper and lower panels in *Figure 5C*,

respectively. These two networks are interconvertible by tuning $K_{CA}$ (the binding affinity of protein C to gene A): (near) zero $1/K_{CA}$ indicates the little effect of protein C to protein A, and thus protein A is not affected by protein C, leading to the activator-inhibitor; non-zero $1/K_{CA}$ implies the existence of the inhibition from protein C to protein A, resulting in the network with an activator-inhibitor and a repressilator. For a given oscillation parameter set for the activator-inhibitor, we first set $1/K_{CA}$ to be (near) zero and calculate period, amplitude, and dimensionless correlation time in the presence of extrinsic or intrinsic noise (the first points in *Figure 5D–G*); then we increase $1/K_{CA}$, and calculate the corresponding quantities (i.e., period, amplitude, and dimensionless correlation time, as shown from the second points in *Figure 5D–G*). We can find that the period, amplitude, and dimensionless correlation time for the activator-inhibitor are usually smaller than those with an additional repressilator, and this gap enlarges with increased $1/K_{CA}$, that is, the increased strength of the negative regulation from the additional node C to the inhibitor node A (*Figure 5C*). Therefore, we demonstrate that adding the repressilator to the activator-inhibitor enhances the oscillation accuracy. This is consistent with that C1 and C2 categories exhibit higher robustness than C3 categories. Moreover, the prolonged period and increased amplitude, which are also observed in C1 and C2 categories, may be the reason for such enhancement (*Figure 5D and E*).

## Discussion

It remains the major challenge in biology to understand how living systems perform complex behaviors accurately in the presence of inevitable noise. Instead of studying biological networks case by case, we try to answer whether there exist general network design principles for living systems to execute biological functions by using a bottom-up strategy (*Lim et al., 2013*; *Zhang and Tang, 2019*).

Here, we systematically explore the network design principles for accurate oscillation in both two- and three-node networks. We identify several key motifs that have distinct robustness to noise. The motif —— a repressilator with positive autoregulation behaves better than other motifs present in *Figure 2A* in most cases, especially the activator-inhibitor oscillator; the additional positive autoregulation can improve the robustness. These results are consistent in spite of sources of noise. However, different sources of noise utilize distinct mechanisms to filter noise: the variability of parameters, a type of extrinsic noise, is largely filtered through long period, and the intrinsic noise is buffered by high amplitude.

Interestingly, investigations of three engineered NF-κB signaling circuits partly validate our simulation results. For the negative-feedback loop circuit, if the additional new regulations form a repressilator, low variance of period will occur, but if no repressilator emerges, the variance of period shows no significant change. These findings show the advantage of the repressilator against noise.

While modifying network topology and changing regulation strength for a fixed topology are both options to improve the robustness of accurate oscillation, each network's robustness is an indicator of the probability of this network topology achieving accurate oscillation with varied regulatory strengths (*Figure 2—figure supplement 2*, *Figure 3—figure supplement 4*): the network topology with high robustness tends to show high dimensionless autocorrelation time when varying regulatory strengths, that is, accurate oscillation (first 10 bars in *Figure 2—figure supplement 2* and *Figure 3—figure supplement 4*); the network topology with low robustness displays a bad performance of oscillation accuracy in the whole parameter space (last 10 bars in *Figure 2—figure supplement 2* and *Figure 3—figure supplement 4*). Besides, our work also suggests that tuning network topology is more efficient than changing regulatory strength. This is based on the observations that network topologies with low robustness (last 10 bars in *Figure 2—figure supplement 2* and *Figure 3—figure supplement 4*) cannot have a high oscillation accuracy even when searching all kinetic parameter space, but changing topologies may increase the probability of high oscillation accuracy. So we suggest that a feasible way to improve the oscillation accuracy in synthetic networks is to first modify the topology to avoid low-robustness ones and then tune the regulation strength, as illustrated in *Figure 5C*.

Mechanisms to buffer different sources of noise in the oscillator can be dramatically different. On the one hand, long period is able to attenuate extrinsic noise, which is also called the time-averaging strategy. This strategy has been widely studied in nonoscillatory networks, such as circuits that are sensitive to the stimulus (*Hornung and Barkai, 2008*), circuits with "switch-like" behaviors (*Wang*

*et al., 2010*), and adaptive circuits (*Nie et al., 2020*; *Sartori and Tu, 2011*). For these nonoscillatory circuits, fluctuations in output have been proven to be related to some key timescales, and long timescales often result in the output with small variance. On the other hand, the intrinsic noise is hard to be attenuated through time averaging, such as the adaptive incoherent feed-forward loop (*Sartori and Tu, 2011*). Actually, the right way to buffer intrinsic noise in biological oscillators was found to depend on levels of molecules. For example, the importance of protein numbers has also been demonstrated in the work of *Potvin-Trottier et al., 2016*. They found that increasing the peak and bottom values decreases the CV in the decay phase of the oscillator. Based on these results, it is suggestive that the network topology with long period and high amplitude may enable good robustness to both extrinsic and intrinsic noise. Interestingly, it is usually not hard to obtain long period and high amplitude simultaneously since the long period tends to allow the protein number to climb to a high level (*Figure 4—figure supplement 1*).

Our work only focused on the effects of biological noise on oscillation accuracy, neglecting other dynamic changes caused by noise. These dynamics may include the loss of multistability and occurrence of oscillation. Specifically, the way to model the noise may cause the loss of multistability (*Duncan et al., 2015*; *Vellela and Qian, 2009*); the presence of noise can produce oscillation even when the corresponding deterministic model cannot oscillate, which has been validated in the toggle-switch system and excitable system (*Lindner et al., 2004*; *Terebus et al., 2019*; *Zaks et al., 2005*). The possible reason might be the noise-induced transition between different states. Since our work only studied network topologies whose deterministic model can generate oscillation, we did not count the topologies that cannot oscillate in the deterministic model but begin to oscillate in the stochastic model. Due to the popularity of such topologies, how these topologies buffer noise will be of interest and may lead to the discoveries of new principles.

In this work, the extrinsic noise is assumed only from fluctuations in kinetic parameters, and its magnitude linearly depends on the level of the parameter. Except this type of extrinsic noise, cells also face the random partitioning that occurs during cell division, noisy stimulus, and so on (*Monti et al., 2018*; *Veliz-Cuba et al., 2015*). Since two different types of noise studied in this work require different mechanisms to buffer, other sources of noise may also need new mechanisms to filter. Thus, some unknown principles need to be further revealed and incorporated into network design as the increasingly improved complexity and multiple sources of noise.

Another limitation of our work is that we did not decompose the reactions in the deterministic model into detailed elementary reaction steps when using the Gillespie algorithm. The advantage of simulating nonelementary reactions with Hill-type rate functions is the low computation cost, and in some biological networks, it leads to consistent results with the model composed of all elementary reactions (*Gonze et al., 2002b*; *Kim et al., 2014*; *Sanft et al., 2011*). However, this approach may not be always accurate, depending on the timescale separation of reactions (*Kim et al., 2014*; *Sanft et al., 2011*); for example, the Hill-type reaction rate is based on the quasi-steady-state approximation, which does not hold when binding/unbinding of TF to the promoter is slow or comparable to the timescales of protein production or decay (*Choi et al., 2008*). Furthermore, this method neglects detailed reaction in gene regulatory networks, and thus fails to study the roles of these reactions in stochasticity. These detailed reactions include the binding and unbinding of the transcription factor to the promoter, dimerization of transcription factors, transcription, and translation (*Cao et al., 2018*; *Shahrezaei and Swain, 2008*; *Terebus et al., 2019*). We anticipate the need for a more detailed model where every reaction of Hill-type form is decomposed into the elementary reactions. The recent development about stochastic algorithms with fast computation makes it feasible to simulate such detailed model for all two- and three-node network topologies, for example, algorithms focusing on solving the chemical master equations (*Cao et al., 2010*; *Cao et al., 2016*; *Munsky and Khammash, 2006*; *Terebus et al., 2021*) and variants of the Gillespie algorithms that directly simulate the temporal dynamics (*Gillespie and Petzold, 2003*). Besides, the construction of probability surfaces through these algorithms may shed light on new principles for accurate oscillation.

# Methods
## Mathematical modeling
### Deterministic model

To model two-node and three-node network topologies, we use transcriptional regulatory networks and assume competitive inhibition among multiple transcription factors. The competitive inhibition means that multiple transcription factors compete for the same binding sites if they regulate one gene simultaneously (**Shi et al., 2017**). So the gene expression depends on the relative weight of transcription factors inhibiting this gene and that activating this gene. The following set of ordinary equations is used to describe the deterministic dynamics of a three-node transcriptional regulatory network:

$$\begin{cases} \frac{dA}{dt} = k_{basal} + \frac{\sum_i v_{iA}\left(\frac{x_i}{K_{iA}}\right)^3 + \delta_A}{1 + \sum_i \left(\frac{x_i}{K_{iA}}\right)^3 + \sum_i \left(\frac{y_i}{K_{iA}}\right)^3} - r_A A \\[2ex] \frac{dB}{dt} = k_{basal} + \frac{\sum_i v_{iB}\left(\frac{x_i}{K_{iB}}\right)^3 + \delta_B}{1 + \sum_i \left(\frac{x_i}{K_{iB}}\right)^3 + \sum_i \left(\frac{y_i}{K_{iB}}\right)^3} - r_B B \\[2ex] \frac{dC}{dt} = k_{basal} + \frac{\sum_i v_{iC}\left(\frac{x_i}{K_{iC}}\right)^3 + \delta_C}{1 + \sum_i \left(\frac{x_i}{K_{iC}}\right)^3 + \sum_i \left(\frac{y_i}{K_{iC}}\right)^3} - r_C C \end{cases} \tag{2}$$

where $A$, $B$, and $C$ are the concentrations of proteins A, B, and C. $x_i = A$, $B$ or $C$ in each equation denotes the concentration of protein activating the gene, and $y_i = A, B$ or $C$ the concentration of protein inhibiting the gene. The production rate constant $v_{ij}$ represents the maximal production rate of protein $i$ regulated by protein $j$, with $K_{ij}$ binding affinity. If there exist proteins activating gene i, $\delta_i$ is zero; if no protein activates gene i, $\delta_i$ is non-zero and represents the production rate caused by other proteins. $k_{basal}$ is the basal production rate. The equations for the two-node transcriptional regulatory network can be obtained in a similar way.

To provide a better explanation about the nonlinear reaction term in above equations, we took the following case as an example: protein A (i.e., TF) binds to gene B to inhibit the gene expression, and protein B binds to the same site in gene B to activate the gene expression. We assumed that (1) there are three binding sites in gene B, which once protein A (or B) binds to, then B (or A) cannot. The elementary reactions are described as follows:

$$A + G_B \rightleftharpoons G_B \cdot A, \quad A + G_B \cdot A \rightleftharpoons G_B \cdot A_2, \quad A + G_B \cdot A_2 \rightleftharpoons G_B \cdot A_3,$$

$$B + G_B \rightleftharpoons G_B \cdot B, \quad B + G_B \cdot B \rightleftharpoons G_B \cdot B_2, \quad B + G_B \cdot B_2 \rightleftharpoons G_B \cdot B_3,$$

where $G_B$, $A$, $B$ denote gene B, protein A, and protein B, respectively. The dissociation rates ($k_{\text{reverse}}/k_{\text{forward}}$) for these six reactions are $K_1$, $K_2$, $\cdots$, $K_6$. Therefore, the fraction of the gene B at the state $G_B \cdot B_3$ in equilibrium is given by

$$\frac{\frac{1}{K_4 K_5 K_6}B^3}{1 + \frac{1}{K_1}A + \frac{1}{K_1 K_2}A^2 + \frac{1}{K_1 K_2 K_3}A^3 + \frac{1}{K_4}B + \frac{1}{K_4 K_5}B^2 + \frac{1}{K_4 K_5 K_6}B^3}.$$

Furthermore, we assumed that (2) $K_6 \ll K_5, K_4$ and $K_3 \ll K_2, K_1$, so that this fraction can be rewritten as

$$\frac{\frac{1}{K_4 K_5 K_6}B^3}{1 + \frac{1}{K_1 K_2 K_3}A^3 + \frac{1}{K_4 K_5 K_6}B^3}.$$

We further assumed (3) that only gene B staying at the state $G_B \cdot B_3$ can lead to transcription and subsequent translation for protein B, and (4) that the binding/unbinding of TFs to a gene can achieve a rapid equilibrium as TF levels change, and thus the production rate of protein B is modeled as

$$v_{BB} \frac{\frac{1}{K_4 K_5 K_6}B^3}{1 + \frac{1}{K_1 K_2 K_3}A^3 + \frac{1}{K_4 K_5 K_6}B^3},$$

where $v_{BB}$ is the maximal production rate when gene B is bound with three protein B. This form is the same as those in (1) if $K_1 K_2 K_3$ and $K_4 K_5 K_6$ are substituted by $K_{AB}^3$ and $K_{BB}^3$, respectively.

## Stochastic model in the presence of extrinsic noise

To generate extrinsic noise, we perturb each kinetic parameter $v_{ij}, \delta_i, r_i$ by multiplying the sum of 1 and an independent temporal noise term, and obtain a new system described by the following stochastic differential equations:

$$
\begin{cases}
\frac{dA}{dt} = k_{basal} + \frac{\sum_i v_{iA}(1+\varepsilon\eta_{iA})\left(\frac{x_i}{K_{iA}}\right)^3 + \delta_A(1+\varepsilon\xi_A)}{1+\sum_i\left(\frac{x_i}{K_{iA}}\right)^3 + \sum_i\left(\frac{y_i}{K_{iA}}\right)^3} - r_A A\left(1+\varepsilon\zeta_A\right) \\[3mm]
\frac{dB}{dt} = k_{basal} + \frac{\sum_i v_{iB}(1+\varepsilon\eta_{iB})\left(\frac{x_i}{K_{iB}}\right)^3 + \delta_B(1+\varepsilon\xi_B)}{1+\sum_i\left(\frac{x_i}{K_{iB}}\right)^3 + \sum_i\left(\frac{y_i}{K_{iB}}\right)^3} - r_B B\left(1+\varepsilon\zeta_B\right) \\[3mm]
\frac{dC}{dt} = k_{basal} + \frac{\sum_i v_{iC}(1+\varepsilon\eta_{iC})\left(\frac{x_i}{K_{iC}}\right)^3 + \delta_C(1+\varepsilon\xi_C)}{1+\sum_i\left(\frac{x_i}{K_{iC}}\right)^3 + \sum_i\left(\frac{y_i}{K_{iC}}\right)^3} - r_C C\left(1+\varepsilon\zeta_C\right)
\end{cases}
\tag{3}
$$

Here, the control parameter $\varepsilon$ indicates magnitude of perturbation of kinetic parameters, and large $\varepsilon$ represents big perturbation of kinetic parameters. $\eta_{ij}$ , $\xi_i$, $\zeta_i$ are independent noise terms and all modeled by the Ornstein–Uhlenbeck process:

$$
\tau^{noise} dz = -zdt + \sigma dW_t
\tag{4}
$$

where $W_t$ is standard Wiener processes. This equation implies that $z(t)$ has zero mean and variance $\frac{\sigma^2}{2\tau^{noise}}$ .

## Stochastic model in the presence of intrinsic noise

To induce intrinsic noise, we replace the concentration of protein with the number of protein by introducing the cell volume $V$ and assume that production events and degradation events occur independently and randomly. To be precise, we use $X_A$, $X_B$, $X_C$ to denote numbers of proteins A, B, C, respectively, and then we replace $A, B, C$ in **Equation 2** by $X_A/V$, $X_B/V$, $X_C/V$, respectively. Therefore, the dynamics of protein numbers $X_A$, $X_B$, $X_C$ are described by the following reactions:

$$
X_A \xrightarrow{Vk_{basal}+V\left(\sum_i v_{iA}\left(\frac{X_i}{K_{iA}}\right)^3 + \delta_A V^3\right)/\left(V^3+\sum_i\left(\frac{X_i}{K_{iA}}\right)^3+\sum_i\left(\frac{Y_i}{K_{iA}}\right)^3\right)} X_A + 1
$$
$$
X_A \xrightarrow{r_A X_A} X_A - 1
$$

$$
X_B \xrightarrow{Vk_{basal}+V\left(\sum_i v_{iB}\left(\frac{X_i}{K_{iB}}\right)^3 + \delta_B V^3\right)/\left(V^3+\sum_i\left(\frac{X_i}{K_{iB}}\right)^3+\sum_i\left(\frac{Y_i}{K_{iB}}\right)^3\right)} X_B + 1
$$
$$
X_B \xrightarrow{r_B X_B} X_B - 1
$$

$$
X_C \xrightarrow{Vk_{basal}+V\left(\sum_i v_{iC}\left(\frac{X_i}{K_{iC}}\right)^3 + \delta_C V^3\right)/\left(V^3+\sum_i\left(\frac{X_i}{K_{iC}}\right)^3+\sum_i\left(\frac{Y_i}{K_{iC}}\right)^3\right)} X_C + 1
$$
$$
X_C \xrightarrow{r_C X_C} X_C - 1
$$

We used the following two algorithms to simulate the above system.

### Gillespie algorithm

We used the standard Gillespie algorithm to simulate the system. There are six reactions in total (as shown above), and the propensity functions are the reaction rates listed above the arrow. Note that we did not decompose the reactions with the Hill function rate into the elementary reactions; the reaction rate with the Hill function type has also been applied to other discrete stochastic models (**Gonze and Goldbeter, 2006**; **Gonze et al., 2002b**; **Veliz-Cuba et al., 2015**; **Wang et al., 2019**; **Zhao et al., 2021**) and proven to be an accurate approximation for the model composed of all elementary reactions under certain circumstances (**Kim et al., 2014**; **Sanft et al., 2011**). In our simulations, each of the six reactions occurs with the random waiting time, which obeys an exponential distribution with mean of the inverse of the propensity function. For example, the reaction of decreasing protein A by 1 has the propensity function $r_A X_A$, and increasing protein A by 1 corresponds to $k_{basel} + V\left(\Sigma_i v_{iA}\left(\frac{X_i}{K_{iA}}\right)^3 + \delta_A V^3\right)/\left(V^3 + \Sigma_i\left(\frac{X_i}{K_{iA}}\right)^3 + \Sigma_i\left(\frac{Y_i}{K_{iA}}\right)^3\right)$. Once we get one trajectory, we can calculate the autocorrelation time. See **Figure 3—figure supplement 2** and **Figure 3—figure supplement 3** for simulation results.

## Langevin equations

We also used the Langevin equation, a good approximation of this system under certain conditions (**Gillespie, 2000**), to model the system. The corresponding chemical Langevin equations are as follows:

$$
\begin{cases}
dX_A = \left( Vk_{basal} + V\dfrac{\sum_i v_{iA}\left(\frac{X_i}{K_{iA}}\right)^3 + \delta_A V^3}{V^3 + \sum_i \left(\frac{X_i}{K_{iA}}\right)^3 + \sum_i \left(\frac{Y_i}{K_{iA}}\right)^3} - r_A X_A \right) dt + \sqrt{Vk_{basal} + V\dfrac{\sum_i v_{iA}\left(\frac{X_i}{K_{iA}}\right)^3 + \delta_A V^3}{V^3 + \sum_i \left(\frac{X_i}{K_{iA}}\right)^3 + \sum_i \left(\frac{Y_i}{K_{iA}}\right)^3} + r_A X_A}\; dW_t^A \\[4mm]
dX_B = \left( Vk_{basal} + V\dfrac{\sum_i v_{iB}\left(\frac{X_i}{K_{iB}}\right)^3 + \delta_B V^3}{V^3 + \sum_i \left(\frac{X_i}{K_{iB}}\right)^3 + \sum_i \left(\frac{Y_i}{K_{iB}}\right)^3} - r_B X_B \right) dt + \sqrt{Vk_{basal} + V\dfrac{\sum_i v_{iB}\left(\frac{X_i}{K_{iB}}\right)^3 + \delta_B V^3}{V^3 + \sum_i \left(\frac{X_i}{K_{iB}}\right)^3 + \sum_i \left(\frac{Y_i}{K_{iB}}\right)^3} + r_B X_B}\; dW_t^B \\[4mm]
dX_C = \left( Vk_{basal} + V\dfrac{\sum_i v_{iC}\left(\frac{X_i}{K_{iC}}\right)^3 + \delta_C V^3}{V^3 + \sum_i \left(\frac{X_i}{K_{iC}}\right)^3 + \sum_i \left(\frac{Y_i}{K_{iC}}\right)^3} - r_C X_C \right) dt + \sqrt{Vk_{basal} + V\dfrac{\sum_i v_{iC}\left(\frac{X_i}{K_{iC}}\right)^3 + \delta_C V^3}{V^3 + \sum_i \left(\frac{X_i}{K_{iC}}\right)^3 + \sum_i \left(\frac{Y_i}{K_{iC}}\right)^3} + r_C X_C}\; dW_t^C
\end{cases}
$$

where $X_i$ is the number of protein $i$, and $W_t^i$ is the standard Wiener process. The control parameter $V$ reflects the magnitude of stochasticity of biological reactions. The big $V$ indicates small degree of stochasticity of biological reactions. See the next section for settings in the numerical simulation.

## Numerical simulations

Numerical simulations for deterministic models were carried out in MATLAB (see https://github.com/LingxiaQiao/oscillation, (copy archived at swh:1:rev:72a2d3d1146b14e7988c1cc06208fe1252e9a6f5; **Qiao, 2022**) for MATLAB scripts). We use the solver ode15s to simulate the dynamics. Simulations for stochastic models were also implemented in MATLAB. In the presence of extrinsic noise, we used the Milstein scheme (**Kloeden and Platen, 2013**) to numerically solve the noise term $\eta_{ij}, \xi_j, \zeta_j$ and used the Euler scheme to solve the dynamics of proteins' concentrations. To be specific, the noise term $z$ ($z = \eta_{ij}, \xi_j, \zeta_j$) at $n+1$ time step is determined by the following manner ($\tau^{noise} = 1$):

$$
z^{(n+1)} = z^{(n)} - z^{(n)}\Delta t + \sigma \delta W_n + \tfrac{1}{2}\sigma^2 \left[ \left(\delta W_n\right)^2 - \Delta t \right]
$$

where $\Delta t$ is the time step, and $\delta W_n$ obeys the normal distribution with mean zero and variance $\Delta t$. Then, the protein's concentration is solved by the Euler scheme (taking A as an example):

$$
[A]^{n+1} = [A]^n + dt\left( k_{basal} + \frac{\sum_i v_{iA}\left(1 + \varepsilon \eta_{iA}^{(n)}\right)\left(\frac{x_i^{(n)}}{K_{iA}}\right)^3 + \delta_A\left(1 + \varepsilon \xi_A^{(n)}\right)}{1 + \sum_i \left(\frac{x_i^{(n)}}{K_{iA}}\right)^3 + \sum_i \left(\frac{y_i^{(n)}}{K_{iA}}\right)^3} - r_A\left(1 + \varepsilon \zeta_A^{(n)}\right) A^{(n)} \right).
$$

In the presence of intrinsic noise, we also used the Milstein scheme to numerically solve the dynamics of proteins' copy numbers. Taking $X_A$ as an example, its value at $n+1$ time step is as follows:

$$
X_A^{(n+1)} = X_A^{(n)} + \left( Vk_{basal} + V\frac{\sum_i v_{iA}\left(\frac{x_i^{(n)}}{K_{iA}}\right)^3 + \delta_A V^3}{V^3 + \sum_i \left(\frac{x_i^{(n)}}{K_{iA}}\right)^3 + \sum_i \left(\frac{y_i^{(n)}}{K_{iA}}\right)^3} - r_A X_A^{(n)} \right)\Delta t + \sigma_A \delta W_A^{(n)} +
$$

$$
\tfrac{1}{2}\sigma_A^2 \left[ \left(\delta W_A^{(n)}\right)^2 - \Delta t \right],
$$

where

$$
\sigma_A = \sqrt{Vk_{basal} + V\frac{\sum_i v_{iA}\left(\frac{X_i^{(n)}}{K_{iA}}\right)^3 + \delta_A V^3}{V^3 + \sum_i \left(\frac{x_i^{(n)}}{K_{iA}}\right)^3 + \sum_i \left(\frac{Y_i^{(n)}}{K_{iA}}\right)^3} + r_A X_A^{(n)}}
$$

## Searching for topologies capable of accurate oscillation

There are two steps for searching for topologies robustly capable of accurate oscillation. The first step is to select network topologies that are able to robustly oscillate among all two- and three-node network topologies. For each topology, 10,000 sets of kinetic parameters are assigned randomly,

with ranges shown in *Supplementary file 1a*; for each parameter set, the initial state of the protein concentration is set to be 0, and we use ode15s in MATLAB to simulate the deterministic dynamics in the time interval [0, 1000]. The dynamics is regarded as oscillation if the following two requirements are satisfied: every protein concentration cannot maintain unchanged in the time interval [700, 1000]; peaks in the time interval [700, 1000] cannot differ a lot. The first requirement excludes the dynamics reaching the steady state, and the second the damping oscillator. We record the number of oscillatory dynamics for each topology, and then regard the topology with this number larger than 80 as the oscillatory topology. But for the topology with the repressilator, if the number of oscillatory dynamics exceeds 10, we still regard this topology as the oscillatory topology. This loose threshold ensures enough oscillatory topologies with the repressilator. In this way, we get 474 oscillatory topologies.

The second step is to explore the robustness of accurate oscillation for the above 474 oscillatory topologies. For each oscillatory topology, we sample enough parameter sets until there are 1000 parameter sets capable of oscillation. For each of these 1000 parameter sets, we record the period T and the amplitude (the maximal peak value among all protein concentrations) from deterministic behavior; next we simulate the stochastic behavior in the time interval [0, 100T]. In the presence of only extrinsic noise, the initial concentration is set to be the state when the concentration B reaches the highest value in a period, but in the presence of intrinsic noise, the initial concentration is converted to the copy number by multiplying the concentration with the cell volume $V$. We use schemes mentioned in the previous sections to numerically solve the stochastic dynamics, with the time step in *Supplementary file 1a*. For a given noisy trajectory, the dimensionless autocorrelation time $\tau/T$ is $-1/\log(c)$, where $c$ is the autocorrelation coefficient at T. Since there are two or three trajectories each of which corresponds to a type of protein, so there are two or three dimensionless autocorrelation times, and we use the largest one as the final dimensionless autocorrelation time. Finally, we use the 90 percentile of dimensionless autocorrelation time to measure the robustness of this topology against noise. The 90 percentile is averaged over five repeated simulations.

## Analytical results for the relation between robustness and period when tuning the timescale *M*

### Phase noise in Demir et al.'s study

In this section, we briefly summarize *Demir et al., 2000* study about the phase noise. We consider the dynamics described by the following equations:

$$\dot{x} = f(x) + B(x)\xi(t) \tag{5}$$

where $x \in \mathbb{R}^3$, $f(\cdot) : \mathbb{R}^3 \to \mathbb{R}^3$, $B(\cdot) : \mathbb{R}^3 \to \mathbb{R}^{3\times3}$ and $\xi(t) \in \mathbb{R}^3$ is random noise. Note that the noise amplitude $B$ is only related to $x$, which is not affected by the time $t$. The unperturbed system $\dot{x} = f(x)$ has a periodic solution $x_s(t)$ (with period T). Linearizing the noise-perturbed system around $x_s(t)$ gives the following system:

$$\dot{w} = A(t)w(t) + B(x_s(t))\xi(t)$$

where $w(t) = x(t) - x_s(t)$, $A(t) = \frac{\partial f(x)}{\partial x}|_{x=x_s(t)}$ is $T$–periodic. From Floquet theory, the state transition matrix $\Phi(t, s)$ for $\dot{w} = A(t)w(t)$ is given by

$$\Phi(t, s) = U(t) \exp(D(t - s)) V(s) = \int_{i=1}^{3} u_i(t) \exp(\mu_i(t - s)) v_i^T(s) \tag{6}$$

where $U(t)$ is $T$-periodic nonsingular matrix with columns denoted by $u_i(t)$, and $V(t)$ with rows denoted by $v_i^T(t)$ is $U^{-1}(t)$. The $\mu_i$'s are Floquet exponents. We can further choose $\mu_1 = 0$ and $u_1(t) = \dot{x}_s(t)$. Then corresponding $v_1(t)$ will play an important role in calculating the phase noise.

From the nonlinear perturbation analysis, $z(t) = x_s(t + \alpha(t)) + y(t)$ solves *Equation 5* for a small $y(t)$. The $\alpha(t)$ and $y(t)$ are called as phase noise and deviation noise, respectively. It can be proved that the variance of the phase noise $\alpha(t)$ increases linearly with time $t$, that is,

$$\text{Var}(\alpha(t)) = ct \tag{7}$$

where $c$ is given by

$$c = \frac{1}{T} \int_0^T v_1^T \left(t'\right) B \left(x_s \left(t'\right)\right) B^T \left(x_s \left(t'\right)\right) v_1 \left(t'\right) dt' . \tag{8}$$

Since $c$ has the same unit as $T$, we divide $c$ by $T$ to ensure a dimensionless index when measuring the phase noise.

## The vector $v_1(t)$ when tuning $M$

We first consider the oscillator governed by the following equation:

$$\dot{x} = f(x) \tag{9}$$

We still use notations in the previous section to denote the quantities for this system. For example, the solution $x_s(t)$, the period T, the Jacobi $A(t)$ of $f(x)$ at the solution $x_s(t)$. We assume the state transition matrix $\Phi(t,s) = U(t) \exp(D(t-s)) V(s)$, which satisfies the first Floquet exponent is 0 and the first column of $U(t)$ is the time derivative of $x_s(t)$. Then the first row of $V(t)$ is denoted as $v_1^T(t)$, which can be used to calculate the variance of phase noise.

Next, we explore how $v_1(t)$ changes when the right-hand term is divided by the timescale $M$. By this way, we obtain

$$\dot{x} = f(x) \frac{1}{M} \tag{10}$$

It is easy to verify that the system governed by Equation 10 has a periodic solution $x_s(t/M)$ with period $MT$. The linearization of this system gives

$$\dot{w} = \frac{1}{M} A \left(\frac{t}{M}\right) w(t)$$

where $w(t) = x(t) - x_s(t/M)$. According to the definition of $\Phi(t,s)$, $\Phi(t,s)$ satisfies

$$\frac{d\Phi(t,s)}{dt} = A(t) \Phi(t,s), \quad \Phi(s,s) = I.$$

So $\Phi\left(\frac{t}{M}, \frac{s}{M}\right)$ satisfies

$$\frac{d\Phi\left(\frac{t}{M},\frac{s}{M}\right)}{dt} = \frac{1}{M} A \left(\frac{t}{M}\right) \Phi \left(\frac{t}{M}, \frac{s}{M}\right), \quad \Phi \left(\frac{s}{M}, \frac{s}{M}\right) = I.$$

Therefore, $\Phi\left(\frac{t}{M}, \frac{s}{M}\right)$ is the state transition matrix for $\dot{w} = \frac{1}{M} A \left(\frac{t}{M}\right) w(t)$. Since $\Phi(t,s) = U(t) \exp(D(t-s)) V(s)$, we can obtain

$$\Phi \left(\frac{t}{M}, \frac{s}{M}\right) = \frac{1}{M} U \left(\frac{t}{M}\right) \exp \left(\frac{D(t-s)}{M}\right) MV \left(\frac{s}{M}\right)$$

where the first term $\frac{1}{M}$ in the right-hand term is to ensure the first column of $\frac{1}{M} U\left(\frac{t}{M}\right)$ is the time derivative of $x_s\left(\frac{t}{M}\right)$, that is, $\frac{1}{M}\dot{x}_s\left(\frac{t}{M}\right)$. Thus, the first row of $MV\left(\frac{t}{M}\right)$ is $Mv_1^T\left(\frac{t}{M}\right)$, which can be used to calculate the variance of phase noise.

## Oscillation accuracy against extrinsic noise when tuning the time scale

For the system governed by Equation 2, we add $\frac{1}{M}$ to the right-hand side and perturb the $v_{ij}, \delta_i, r_i$ to introduce the extrinsic noise, thus leading to the following equations:

$$\begin{cases} \frac{dA}{dt} = \left(k_{basal} + \frac{\sum_i v_{iA}\left(\frac{x_i}{K_{iA}}\right)^3 + \delta_A}{1+\sum_i \left(\frac{x_i}{K_{iA}}\right)^3 + \sum_i \left(\frac{y_i}{K_{iA}}\right)^3} - r_A A\right) \frac{1}{M} + \varepsilon \frac{1}{M} \left(\frac{\sum_i v_{iA}\eta_{iA}\left(\frac{x_i}{K_{iA}}\right)^3 + \delta_A \xi_A}{1+\sum_i \left(\frac{x_i}{K_{iA}}\right)^3 + \sum_i \left(\frac{y_i}{K_{iA}}\right)^3} - r_A A \zeta_A\right) \\ \frac{dB}{dt} = \left(k_{basal} + \frac{\sum_i v_{iB}\left(\frac{x_i}{K_{iB}}\right)^3 + \delta_B}{1+\sum_i \left(\frac{x_i}{K_{iB}}\right)^3 + \sum_i \left(\frac{y_i}{K_{iB}}\right)^3} - r_B B\right) \frac{1}{M} + \varepsilon \frac{1}{M} \left(\frac{\sum_i v_{iB}\eta_{iB}\left(\frac{x_i}{K_{iB}}\right)^3 + \delta_B \xi_B}{1+\sum_i \left(\frac{x_i}{K_{iB}}\right)^3 + \sum_i \left(\frac{y_i}{K_{iB}}\right)^3} - r_B B \zeta_B\right) \\ \frac{dC}{dt} = \left(k_{basal} + \frac{\sum_i v_{iC}\left(\frac{x_i}{K_{iC}}\right)^3 + \delta_C}{1+\sum_i \left(\frac{x_i}{K_{iC}}\right)^3 + \sum_i \left(\frac{y_i}{K_{iC}}\right)^3} - r_C C\right) \frac{1}{M} + \varepsilon \frac{1}{M} \left(\frac{\sum_i v_{iC}\eta_{iC}\left(\frac{x_i}{K_{iC}}\right)^3 + \delta_C \xi_C}{1+\sum_i \left(\frac{x_i}{K_{iC}}\right)^3 + \sum_i \left(\frac{y_i}{K_{iC}}\right)^3} - r_C C \zeta_C\right) \end{cases},$$

For simplicity, we still use $x$ and $f(x)$ to denote $(A, B, C)$ and the terms in the first brackets in the right-hand terms, respectively. Thus, the above equations can be rewritten as

$$\dot{x} = f(x)\frac{1}{M} + \frac{1}{M}B_{ex}(x)\Pi(t)$$

where $\Pi(t) = (\eta_{AA}, \eta_{BA}, \eta_{CA}, \xi_A, \zeta_A, \eta_{AB}, \eta_{BB}, \eta_{CB}, \xi_B, \zeta_B, \eta_{AC}, \eta_{BC}, \eta_{CC}, \xi_C, \zeta_C)$, and $B_{ex}(x)$ is the matrix whose elements are coefficients of the random noise when $M = 1$. Recall that the ' $v_1(t)$ ' in *Equation 8* is $Mv_1\left(\frac{t}{M}\right)$ for the system $\dot{x} = f(x)\frac{1}{M}$, so the slope of variance of phase noise over period for the system with timescale $M$ is

$$\frac{1}{MT}\frac{1}{MT}\int_0^{MT} Mv_1^T\left(\frac{t'}{M}\right)\frac{1}{M}B_{ex}\left(x_s\left(t'/M\right)\right)\frac{1}{M}B_{ex}^T\left(x_s\left(t'/M\right)\right)Mv_1\left(\frac{t'}{M}\right)dt'$$

By replacing $t'$ with $t''M$ and using $t'$ to denote $t''$, we obtain

$$\frac{1}{MT^2}\int_0^T v_1^T\left(t'\right)B\left(x_s\left(t'\right)\right)B^T\left(x_s\left(t'\right)\right)v_1\left(t'\right)dt'$$

So it can be concluded that large $M$ causes small normalized phase noise in the presence of extrinsic noise. As large $M$ also leads to long period but has no effect on the amplitude, the long period might be the reason for high oscillation accuracy in the presence of extrinsic noise.

## Oscillation accuracy against intrinsic noise when tuning the timescale

Similarly, for the system governed by Equation 2, we add $\frac{1}{M}$ to the right-hand side and introduce the cell volume $V$ to incorporate the intrinsic noise, thus leading to the following chemical Langevin equations:

$$\begin{cases} dX_A = \left(Vk_{basal} + V\dfrac{\sum_i v_{iA}\left(\frac{X_i}{K_{iA}}\right)^3 + \delta_A V^3}{V^3 + \sum_i\left(\frac{X_i}{K_{iA}}\right)^3 + \sum_i\left(\frac{Y_i}{K_{iA}}\right)^3} - r_A X_A\right)\frac{1}{M}dt + \sqrt{\frac{1}{M}}\sqrt{Vk_{basal} + V\dfrac{\sum_i v_{iA}\left(\frac{X_i}{K_{iA}}\right)^3 + \delta_A V^3}{V^3 + \sum_i\left(\frac{X_i}{K_{iA}}\right)^3 + \sum_i\left(\frac{Y_i}{K_{iA}}\right)^3} + r_A X_A}\,dW_t^A \\[2em] dX_B = \left(Vk_{basal} + V\dfrac{\sum_i v_{iB}\left(\frac{X_i}{K_{iB}}\right)^3 + \delta_B V^3}{V^3 + \sum_i\left(\frac{X_i}{K_{iB}}\right)^3 + \sum_i\left(\frac{Y_i}{K_{iB}}\right)^3} - r_B X_B\right)\frac{1}{M}dt + \sqrt{\frac{1}{M}}\sqrt{Vk_{basal} + V\dfrac{\sum_i v_{iB}\left(\frac{X_i}{K_{iB}}\right)^3 + \delta_B V^3}{V^3 + \sum_i\left(\frac{X_i}{K_{iB}}\right)^3 + \sum_i\left(\frac{Y_i}{K_{iB}}\right)^3} + r_B X_B}\,dW_t^B \\[2em] dX_C = \left(Vk_{basal} + V\dfrac{\sum_i v_{iC}\left(\frac{X_i}{K_{iC}}\right)^3 + \delta_C V^3}{V^3 + \sum_i\left(\frac{X_i}{K_{iC}}\right)^3 + \sum_i\left(\frac{Y_i}{K_{iC}}\right)^3} - r_C X_C\right)\frac{1}{M}dt + \sqrt{\frac{1}{M}}\sqrt{Vk_{basal} + V\dfrac{\sum_i v_{iC}\left(\frac{X_i}{K_{iC}}\right)^3 + \delta_C V^3}{V^3 + \sum_i\left(\frac{X_i}{K_{iC}}\right)^3 + \sum_i\left(\frac{Y_i}{K_{iC}}\right)^3} + r_C X_C}\,dW_t^C \end{cases}$$

We use $X$ and $f(X)$ to denote $(X_A, X_B, X_C)$ and the terms in the first brackets in the right-hand terms, respectively. Thus, the above equations can be rewritten as

$$\dot{X} = f(X)\frac{1}{M} + \sqrt{\frac{1}{M}}B_{in}(X)\Lambda(t)$$

where $\Lambda(t)$ is $(dW_A/dt, dW_B/dt, dW_C/dt)$, and $B_{in}(X)$ is the matrix whose elements are coefficients of the random noise when $M = 1$. If we use $v_1^{in}(t)$ to represent the ' $v_1(t)$ ' for the system $\dot{X} = f(X)$, the ' $v_1(t)$ ' for the system $\dot{X} = f(X)\frac{1}{M}$ is $Mv_1^{in}\left(\frac{t}{M}\right)$. So the slope of variance of phase noise over period for the system with cell volume $V$ is

$$\frac{1}{MT}\frac{1}{MT}\int_0^{MT} M\left(v_1^{in}\left(\frac{t'}{M}\right)\right)^T\sqrt{\frac{1}{M}}B_{in}\left(x_s\left(t'/M\right)\right)\sqrt{\frac{1}{M}}B_{in}^T\left(x_s\left(t'/M\right)\right)Mv_1^{in}\left(\frac{t'}{M}\right)dt'$$

By replacing $t'$ with $t''M$ and using $t'$ to denote $t''$, we obtain

$$\frac{1}{T^2}\int_0^T \left(v_1^{in}\left(t'\right)\right)^T B\left(x_s\left(t'\right)\right)B^T\left(x_s\left(t'\right)\right)v_1^{in}\left(t'\right)dt'$$

It can be seen that $M$ has no effect on the normalized slope of variance of phase noise, so the long period might not influence the noise in proteins when facing intrinsic noise.

## Analytical results for the relation between robustness and amplitude when tuning the rescaling parameter $N$

### Deterministic model with rescaled variables

To analyze the relation between amplitude and oscillation accuracy against noise, we replace $(A,\ B,C)$ in **Equation 2** with $\left(\widetilde{A}/N,\ \widetilde{B}/N,\widetilde{C}/N\right)$, which allows us to tune the amplitude by varying $N$. After this rescaling, we obtain the following equations for $\widetilde{A}$ , $\widetilde{B}$ and $\widetilde{C}$:

$$
\begin{cases}
\dfrac{d\widetilde{A}}{dt} = Nk_{basal} + \dfrac{\sum_i N v_{iA}\left(\frac{\widetilde{x}_i}{NK_{iA}}\right)^3 + N\delta_A}{1+\sum_i\left(\frac{\widetilde{x}_i}{NK_{iA}}\right)^3 + \sum_i\left(\frac{\widetilde{y}_i}{NK_{iA}}\right)^3} - r_A\widetilde{A} \\[3em]
\dfrac{d\widetilde{B}}{dt} = Nk_{basal} + \dfrac{\sum_i N v_{iB}\left(\frac{\widetilde{x}_i}{NK_{iB}}\right)^3 + N\delta_B}{1+\sum_i\left(\frac{\widetilde{x}_i}{NK_{iB}}\right)^3 + \sum_i\left(\frac{\widetilde{y}_i}{NK_{iB}}\right)^3} - r_B\widetilde{B} \\[3em]
\dfrac{d\widetilde{C}}{dt} = Nk_{basal} + \dfrac{\sum_i N v_{iC}\left(\frac{\widetilde{x}_i}{NK_{iC}}\right)^3 + N\delta_C}{1+\sum_i\left(\frac{\widetilde{x}_i}{NK_{iC}}\right)^3 + \sum_i\left(\frac{\widetilde{y}_i}{NK_{iC}}\right)^3} - r_C\widetilde{C}
\end{cases}
\tag{11}
$$

where $\widetilde{x}_i = \frac{x_i}{N}$, $\widetilde{y}_i = \frac{y_i}{N}$ . If $N=1$, this equation is the same as **Equation 2**, so $\widetilde{A}$ , $\widetilde{B}$, and $\widetilde{C}$ show same amplitudes with $A$, $B$, and $C$, respectively. Nevertheless, if $N\neq 1$, the amplitude of $\widetilde{A}$ , $\widetilde{B}$, or $\widetilde{C}$ is $N$ times as high as that of $A$, $B$, or $C$. Note that $N$ has no effect on the period.

### Oscillation accuracy against extrinsic noise when tuning the rescaling parameter

In the system governed by Equation 11, we assume that $N$ causes binding affinities, $v_{ij}$ 's, and $\delta_i$ 's to $N$ times their original values, but the $r_i$ 's remain unchanged. Next, we consider the system described by Equation 11 in the presence of only extrinsic noise. We perturb each kinetic parameter $v_{ij}, \delta_i, r_i$ by the same method as mentioned in the section 'Mathematical modeling' and obtain a new system described by the following equations:

$$
\begin{cases}
\dfrac{d\widetilde{A}}{dt} = Nk_{basal} + \dfrac{\sum_i N v_{iA}\left(\frac{\widetilde{x}_i}{NK_{iA}}\right)^3 + N\delta_A}{1+\sum_i\left(\frac{\widetilde{x}_i}{NK_{iA}}\right)^3 + \sum_i\left(\frac{\widetilde{y}_i}{NK_{iA}}\right)^3} - r_A\widetilde{A} + \varepsilon\left(\dfrac{\sum_i N v_{iA}\eta_{iA}\left(\frac{\widetilde{x}_i}{NK_{iA}}\right)^3 + N\delta_A\xi_A}{1+\sum_i\left(\frac{\widetilde{x}_i}{NK_{iA}}\right)^3 + \sum_i\left(\frac{\widetilde{y}_i}{NK_{iA}}\right)^3} - r_A\widetilde{A}\zeta_A\right) \\[3em]
\dfrac{d\widetilde{B}}{dt} = Nk_{basal} + \dfrac{\sum_i N v_{iB}\left(\frac{\widetilde{x}_i}{NK_{iB}}\right)^3 + N\delta_B}{1+\sum_i\left(\frac{\widetilde{x}_i}{NK_{iB}}\right)^3 + \sum_i\left(\frac{\widetilde{y}_i}{NK_{iB}}\right)^3} - r_B\widetilde{B} + \varepsilon\left(\dfrac{\sum_i N v_{iB}\eta_{iA}\left(\frac{\widetilde{x}_i}{NK_{iB}}\right)^3 + N\delta_B\xi_A}{1+\sum_i\left(\frac{\widetilde{x}_i}{NK_{iB}}\right)^3 + \sum_i\left(\frac{\widetilde{y}_i}{NK_{iB}}\right)^3} - r_B\widetilde{B}\zeta_A\right) \\[3em]
\dfrac{d\widetilde{C}}{dt} = Nk_{basal} + \dfrac{\sum_i N v_{iC}\left(\frac{\widetilde{x}_i}{NK_{iC}}\right)^3 + N\delta_C}{1+\sum_i\left(\frac{\widetilde{x}_i}{NK_{iC}}\right)^3 + \sum_i\left(\frac{\widetilde{y}_i}{NK_{iC}}\right)^3} - r_C\widetilde{C} + \varepsilon\left(\dfrac{\sum_i N v_{iC}\eta_{iA}\left(\frac{\widetilde{x}_i}{NK_{iC}}\right)^3 + N\delta_C\xi_A}{1+\sum_i\left(\frac{\widetilde{x}_i}{NK_{iC}}\right)^3 + \sum_i\left(\frac{\widetilde{y}_i}{NK_{iC}}\right)^3} - r_C\widetilde{C}\zeta_A\right)
\end{cases}
$$

where $\eta_{ij}$ , $\xi_i, \zeta_i$ are independent noise terms and all modeled by **Equation 4** By multiplying $1/N$ to both sides of above equations, we get

$$
\begin{cases}
\dfrac{1}{N}\dfrac{d\widetilde{A}}{dt} = k_{basal} + \dfrac{\sum_i v_{iA}\left(\frac{\widetilde{x}_i}{NK_{iA}}\right)^3 + \delta_A}{1+\sum_i\left(\frac{\widetilde{x}_i}{NK_{iA}}\right)^3 + \sum_i\left(\frac{\widetilde{y}_i}{NK_{iA}}\right)^3} - r_A\dfrac{1}{N}\widetilde{A} + \varepsilon\left(\dfrac{\sum_i v_{iA}\eta_{iA}\left(\frac{\widetilde{x}_i}{NK_{iA}}\right)^3 + \delta_A\xi_A}{1+\sum_i\left(\frac{\widetilde{x}_i}{NK_{iA}}\right)^3 + \sum_i\left(\frac{\widetilde{y}_i}{NK_{iA}}\right)^3} - r_A\dfrac{1}{N}\widetilde{A}\zeta_A\right) \\[3em]
\dfrac{1}{N}\dfrac{d\widetilde{B}}{dt} = k_{basal} + \dfrac{\sum_i v_{iB}\left(\frac{\widetilde{x}_i}{NK_{iB}}\right)^3 + \delta_B}{1+\sum_i\left(\frac{\widetilde{x}_i}{NK_{iB}}\right)^3 + \sum_i\left(\frac{\widetilde{y}_i}{NK_{iB}}\right)^3} - r_B\dfrac{1}{N}\widetilde{B} + \varepsilon\left(\dfrac{\sum_i v_{iB}\eta_{iA}\left(\frac{\widetilde{x}_i}{NK_{iB}}\right)^3 + \delta_B\xi_A}{1+\sum_i\left(\frac{\widetilde{x}_i}{NK_{iB}}\right)^3 + \sum_i\left(\frac{\widetilde{y}_i}{NK_{iB}}\right)^3} - r_B\dfrac{1}{N}\widetilde{B}\zeta_A\right) \\[3em]
\dfrac{1}{N}\dfrac{d\widetilde{C}}{dt} = k_{basal} + \dfrac{\sum_i v_{iC}\left(\frac{\widetilde{x}_i}{NK_{iC}}\right)^3 + \delta_C}{1+\sum_i\left(\frac{\widetilde{x}_i}{NK_{iC}}\right)^3 + \sum_i\left(\frac{\widetilde{y}_i}{NK_{iC}}\right)^3} - r_C\dfrac{1}{N}\widetilde{C} + \varepsilon\left(\dfrac{\sum_i v_{iC}\eta_{iA}\left(\frac{\widetilde{x}_i}{NK_{iC}}\right)^3 + \delta_C\xi_A}{1+\sum_i\left(\frac{\widetilde{x}_i}{NK_{iC}}\right)^3 + \sum_i\left(\frac{\widetilde{y}_i}{NK_{iC}}\right)^3} - r_C\dfrac{1}{N}\widetilde{C}\zeta_A\right)
\end{cases}
$$

Let $\widetilde{\widetilde{A}} = \frac{\widetilde{A}}{N}, \widetilde{\widetilde{B}} = \frac{\widetilde{B}}{N}$, and $\widetilde{\widetilde{B}} = \frac{\widetilde{B}}{N}$, the set of equations for $\widetilde{\widetilde{A}}$, $\widetilde{\widetilde{B}}$, and $\widetilde{\widetilde{C}}$ is the same as that in Equation 3, in which $N$ do not appear. So the dynamics of $\widetilde{\widetilde{A}}$, $\widetilde{\widetilde{B}}$, or $\widetilde{\widetilde{C}}$ will not change with varied $N$, thus leading to the same accuracy of oscillation when varying $N$. Based on $\widetilde{\widetilde{A}}N = \widetilde{A}, \widetilde{\widetilde{B}}N = \widetilde{B}$, and $\widetilde{\widetilde{C}}N = \widetilde{C}$ and the fact that the rescaling has no effect on the correlation function, $\widetilde{A}$ shows the same accuracy of oscillation as $\widetilde{\widetilde{A}}$, and so does $\widetilde{B}$ or $\widetilde{C}$. Therefore, in the system for $\widetilde{A}$, $\widetilde{B}$, and $\widetilde{C}$, its oscillation accuracy remains the same with varied $N$. Since $N$ influences the amplitude while maintaining the period, the change in the amplitude will not affect the oscillation noise against extrinsic noise.

## Oscillation accuracy against intrinsic noise when tuning the rescaling parameter

The dynamics of the system described by Equation 11. in the presence of only intrinsic noise is governed by

$$
\begin{cases}
dX_{\widetilde{A}} = \left( VN k_{basal} + V \dfrac{\sum_i Nv_{iA}\left(\frac{X_{\widetilde{i}}}{NK_{iA}}\right)^3 + N\delta_A V^3}{V^3 + \sum_i \left(\frac{X_{\widetilde{i}}}{NK_{iA}}\right)^3 + \sum_i \left(\frac{Y_{\widetilde{i}}}{NK_{iA}}\right)^3} - r_A X_{\widetilde{A}} \right) dt \\
\quad + \sqrt{ VN k_{basal} + V \dfrac{\sum_i Nv_{iA}\left(\frac{X_{\widetilde{i}}}{NK_{iA}}\right)^3 + N\delta_A V^3}{V^3 + \sum_i \left(\frac{X_{\widetilde{i}}}{NK_{iA}}\right)^3 + \sum_i \left(\frac{Y_{\widetilde{i}}}{NK_{iA}}\right)^3} + r_A X_{\widetilde{A}}} \, dW_t^A \\[4pt]
dX_{\widetilde{B}} = \left( VN k_{basal} + V \dfrac{\sum_i Nv_{iB}\left(\frac{X_{\widetilde{i}}}{NK_{iB}}\right)^3 + N\delta_B V^3}{V^3 + \sum_i \left(\frac{X_{\widetilde{i}}}{NK_{iB}}\right)^3 + \sum_i \left(\frac{Y_{\widetilde{i}}}{NK_{iB}}\right)^3} - r_B X_{\widetilde{B}} \right) dt \\
\quad + \sqrt{ VN k_{basal} + V \dfrac{\sum_i Nv_{iB}\left(\frac{X_{\widetilde{i}}}{NK_{iB}}\right)^3 + N\delta_B V^3}{V^3 + \sum_i \left(\frac{X_{\widetilde{i}}}{NK_{iB}}\right)^3 + \sum_i \left(\frac{Y_{\widetilde{i}}}{NK_{iB}}\right)^3} + r_B X_{\widetilde{B}}} \, dW_t^B \\[4pt]
dX_{\widetilde{C}} = \left( VN k_{basal} + V \dfrac{\sum_i Nv_{iC}\left(\frac{X_{\widetilde{i}}}{NK_{iC}}\right)^3 + N\delta_C V^3}{V^3 + \sum_i \left(\frac{X_{\widetilde{i}}}{NK_{iC}}\right)^3 + \sum_i \left(\frac{Y_{\widetilde{i}}}{NK_{iC}}\right)^3} - r_C X_{\widetilde{C}} \right) dt \\
\quad + \sqrt{ VN k_{basal} + V \dfrac{\sum_i Nv_{iC}\left(\frac{X_{\widetilde{i}}}{NK_{iC}}\right)^3 + N\delta_C V^3}{V^3 + \sum_i \left(\frac{X_{\widetilde{i}}}{NK_{iC}}\right)^3 + \sum_i \left(\frac{Y_{\widetilde{i}}}{NK_{iC}}\right)^3} + r_C X_{\widetilde{C}}} \, dW_t^C
\end{cases}
$$

where $X_{\widetilde{A}} = V\widetilde{A}$, $X_{\widetilde{B}} = V\widetilde{B}$, and $X_{\widetilde{C}} = V\widetilde{C}$, and $V$ is the cell volume. By multiplying $1/N$ to both sides of above equations, we get

$$
\begin{cases}
\frac{1}{N}dX_{\underset{\sim}{A}} = \left( Vk_{basal} + V\frac{\sum_i v_{iA}\left(\frac{X_{\underset{\sim}{i}}}{NK_{iA}}\right)^3 + \delta_A V^3}{V^3 + \sum_i \left(\frac{X_{\underset{\sim}{i}}}{NK_{iA}}\right)^3 + \sum_i \left(\frac{Y_{\underset{\sim}{i}}}{NK_{iA}}\right)^3} - r_A \frac{1}{N}X_{\underset{\sim}{A}} \right) dt \\[3mm]
\quad + \sqrt{\frac{1}{N}} \sqrt{Vk_{basal} + V\frac{\sum_i v_{iA}\left(\frac{X_{\underset{\sim}{i}}}{NK_{iA}}\right)^3 + \delta_A V^3}{V^3 + \sum_i \left(\frac{X_{\underset{\sim}{i}}}{NK_{iA}}\right)^3 + \sum_i \left(\frac{Y_{\underset{\sim}{i}}}{NK_{iA}}\right)^3} + r_A \frac{1}{N}X_{\underset{\sim}{A}}}\, dW_t^A \\[4mm]
\frac{1}{N}dX_{\underset{\sim}{B}} = \left( Vk_{basal} + V\frac{\sum_i v_{iB}\left(\frac{X_{\underset{\sim}{i}}}{NK_{iB}}\right)^3 + \delta_B V^3}{V^3 + \sum_i \left(\frac{X_{\underset{\sim}{i}}}{NK_{iB}}\right)^3 + \sum_i \left(\frac{Y_{\underset{\sim}{i}}}{NK_{iB}}\right)^3} - r_B \frac{1}{N}X_{\underset{\sim}{B}} \right) dt \\[3mm]
\quad + \sqrt{\frac{1}{N}} \sqrt{Vk_{basal} + V\frac{\sum_i v_{iB}\left(\frac{X_{\underset{\sim}{i}}}{NK_{iB}}\right)^3 + \delta_B V^3}{V^3 + \sum_i \left(\frac{X_{\underset{\sim}{i}}}{NK_{iB}}\right)^3 + \sum_i \left(\frac{Y_{\underset{\sim}{i}}}{NK_{iB}}\right)^3} + r_B \frac{1}{N}X_{\underset{\sim}{B}}}\, dW_t^B \\[4mm]
\frac{1}{N}dX_{\underset{\sim}{C}} = \left( Vk_{basal} + V\frac{\sum_i v_{iC}\left(\frac{X_{\underset{\sim}{i}}}{NK_{iC}}\right)^3 + \delta_C V^3}{V^3 + \sum_i \left(\frac{X_{\underset{\sim}{i}}}{NK_{iC}}\right)^3 + \sum_i \left(\frac{Y_{\underset{\sim}{i}}}{NK_{iC}}\right)^3} - r_C \frac{1}{N}X_{\underset{\sim}{C}} \right) dt \\[3mm]
\quad + \sqrt{\frac{1}{N}} \sqrt{Vk_{basal} + V\frac{\sum_i v_{iC}\left(\frac{X_{\underset{\sim}{i}}}{NK_{iC}}\right)^3 + \delta_C V^3}{V^3 + \sum_i \left(\frac{X_{\underset{\sim}{i}}}{NK_{iC}}\right)^3 + \sum_i \left(\frac{Y_{\underset{\sim}{i}}}{NK_{iC}}\right)^3} + r_C \frac{1}{N}X_{\underset{\sim}{C}}}\, dW_t^C
\end{cases}
$$

Let $X_{\underset{\approx}{A}} = \frac{X_{\underset{\sim}{A}}}{N}, X_{\underset{\approx}{B}} = \frac{X_{\underset{\sim}{B}}}{N}$, and $X_{\underset{\approx}{C}} = \frac{X_{\underset{\sim}{C}}}{N}$ , equations for $X_{\underset{\approx}{A}}$ , $X_{\underset{\approx}{B}}$ , and $X_{\underset{\approx}{C}}$ are

$$
\begin{cases}
dX_{\underset{\approx}{A}} = \left( Vk_{basal} + V\frac{\sum_i v_{iA}\left(\frac{X_{\underset{\approx}{i}}}{K_{iA}}\right)^3 + \delta_A V^3}{V^3 + \sum_i \left(\frac{X_{\underset{\approx}{i}}}{K_{iA}}\right)^3 + \sum_i \left(\frac{Y_{\underset{\approx}{i}}}{K_{iA}}\right)^3} - r_A X_{\underset{\approx}{A}} \right) dt \\[3mm]
\quad + \sqrt{\frac{1}{N}} \sqrt{Vk_{basal} + V\frac{\sum_i v_{iA}\left(\frac{X_{\underset{\approx}{i}}}{K_{iA}}\right)^3 + \delta_A V^3}{V^3 + \sum_i \left(\frac{X_{\underset{\approx}{i}}}{K_{iA}}\right)^3 + \sum_i \left(\frac{Y_{\underset{\approx}{i}}}{K_{iA}}\right)^3} + r_A X_{\underset{\approx}{A}}}\, dW_t^A \\[4mm]
dX_{\underset{\approx}{B}} = \left( Vk_{basal} + V\frac{\sum_i v_{iB}\left(\frac{X_{\underset{\approx}{i}}}{K_{iB}}\right)^3 + \delta_B V^3}{V^3 + \sum_i \left(\frac{X_{\underset{\approx}{i}}}{K_{iB}}\right)^3 + \sum_i \left(\frac{Y_{\underset{\approx}{i}}}{K_{iB}}\right)^3} - r_B X_{\underset{\approx}{B}} \right) dt \\[3mm]
\quad + \sqrt{\frac{1}{N}} \sqrt{Vk_{basal} + V\frac{\sum_i v_{iB}\left(\frac{X_{\underset{\approx}{i}}}{K_{iB}}\right)^3 + \delta_B V^3}{V^3 + \sum_i \left(\frac{X_{\underset{\approx}{i}}}{K_{iB}}\right)^3 + \sum_i \left(\frac{Y_{\underset{\approx}{i}}}{K_{iB}}\right)^3} + r_B X_{\underset{\approx}{B}}}\, dW_t^B \\[4mm]
dX_{\underset{\approx}{C}} = \left( Vk_{basal} + V\frac{\sum_i v_{iC}\left(\frac{X_{\underset{\approx}{i}}}{K_{iC}}\right)^3 + \delta_C V^3}{V^3 + \sum_i \left(\frac{X_{\underset{\approx}{i}}}{K_{iC}}\right)^3 + \sum_i \left(\frac{Y_{\underset{\approx}{i}}}{K_{iC}}\right)^3} - r_C X_{\underset{\approx}{C}} \right) dt \\[3mm]
\quad + \sqrt{\frac{1}{N}} \sqrt{Vk_{basal} + V\frac{\sum_i v_{iC}\left(\frac{X_{\underset{\approx}{i}}}{K_{iC}}\right)^3 + \delta_C V^3}{V^3 + \sum_i \left(\frac{X_{\underset{\approx}{i}}}{K_{iC}}\right)^3 + \sum_i \left(\frac{Y_{\underset{\approx}{i}}}{K_{iC}}\right)^3} + r_C X_{\underset{\approx}{C}}}\, dW_t^C
\end{cases}
$$

In above equations, $N$ only negatively affects the magnitude of noise term, so the oscillation accuracies of $X_{\underset{A}{\sim}}$, $X_{\underset{B}{\sim}}$, and $X_{\underset{C}{\sim}}$ increased with increased $N$. Thus, the oscillation accuracies of $X_{\underset{A}{\sim}}$, $X_{\underset{B}{\sim}}$, and $X_{\underset{C}{\sim}}$ also increased with increased $N$ because the correlation function is not affected by the rescaling operation. Besides, large $N$ increase the amplitude while maintaining the period. Taken together, the high amplitude may enhance the oscillation noise against intrinsic noise.

## Acknowledgements

LZ was partly supported by the National Key Research and Development Program of China 2021YFF1200500 and National Natural Science Foundation of China No. 12050002. PW was partly supported by the National Key Basic Research Program of China 2018YFA0902800 and the National Natural Science Foundation of China 31622022.

## Additional information

### Funding

| Funder | Grant reference number | Author |
| --- | --- | --- |
| National Key Research and Development Program of China | 2021YFF1200500 | Lei Zhang |
| National Natural Science Foundation of China | 12050002 | Lei Zhang |
| National Key Basic Research Program of China | 2018YFA0902800 | Ping Wei |
| National Natural Science Foundation of China | 31622022 | Ping Wei |

The funders had no role in study design, data collection and interpretation, or the decision to submit the work for publication.

### Author contributions

Lingxia Qiao, Zhi-Bo Zhang, Wei Zhao, Formal analysis, Validation, Investigation, Methodology, Writing – original draft, Writing – review and editing; Ping Wei, Supervision, Funding acquisition, Project administration, Writing – review and editing; Lei Zhang, Conceptualization, Supervision, Funding acquisition, Methodology, Project administration, Writing – review and editing

### Author ORCIDs

Lingxia Qiao http://orcid.org/0000-0003-2639-6851
Zhi-Bo Zhang http://orcid.org/0000-0002-3316-2634
Ping Wei http://orcid.org/0000-0002-1275-7145
Lei Zhang http://orcid.org/0000-0001-9972-2051

### Decision letter and Author response

Decision letter https://doi.org/10.7554/eLife.76188.sa1
Author response https://doi.org/10.7554/eLife.76188.sa2

## Additional files

### Supplementary files

• Supplementary file 1. Kinetic parameters used in numerical simulations. (a) Parameters for searching topologies. (b) Parameters used in *Figure 4*. (c) Parameters used in *Figure 5*.

• MDAR checklist

## Data availability

The current manuscript is a computational study. Modelling code and NFKB data for plotting are uploaded to GitHub at https://github.com/LingxiaQiao/oscillation, (copy archived at swh:1:rev:72a2d3d1146b14e7988c1cc06208fe1252e9a6f5).

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
