## [Editor Report]

The authors study the important problem of how to achieve accurate oscillation robustly in biological networks where noise level may be high. The authors adopted a comprehensive approach and study how different network configurations affect oscillation. This work makes an important contribution to the field as it offers the first comprehensive survey of networks motifs capable of oscillation, with further characterization of their robustness.

---

## [Decision Letter]

**Decision letter after peer review:**

Thank you for submitting your article "Network design principle for robust oscillatory behaviors with respect to biological noise" for consideration by *eLife*. Your article has been reviewed by 2 peer reviewers, and the evaluation has been overseen by a Reviewing Editor and Naama Barkai as the Senior Editor. The reviewers have opted to remain anonymous.

The reviewers found the work interesting, but reviewer #2 raised some issues that need to be addressed prior to publication. please address all these issues below.

*Reviewer #1 (Recommendations for the authors):*

The authors may consider to discuss further how the regulation changes under the fixed topology influence the oscillation quality compared with those under different topologies and see which one causes more significant changes.

*Reviewer #2 (Recommendations for the authors):*

The initial analysis was based on oscillation behavior of ODE models at different parameter values. There have been detailed studies in the past with this approach, and the current results appear to be similar, eg. oscillation appear to coincide with the graph cycle structures in the network. The authors may wish to discuss their findings from ODE analysis with previous results of 'Glass, Leon, and Joel S. Pasternack. "Stable oscillations in mathematical models of biological control systems." Journal of Mathematical Biology 6.3 (1978): 207-223.’ and explain the relationship of their findings with these earlier results.

---

## [Author Response]

Reviewer #1 (Recommendations for the authors):The authors may consider to discuss further how the regulation changes under the fixed topology influence the oscillation quality compared with those under different topologies and see which one causes more significant changes.

Thank you for the good suggestion. In the revision, we made two new figures (Figure 2—figure supplement 2 and Figure 3—figure supplement 4) and added the following paragraph in the Discussion:

“While modifying network topology and changing regulation strength for a fixed topology are both options to improve the robustness of accurate oscillation, each network’s robustness is an indicator of the probability of this network topology achieving accurate oscillation with varied regulatory strengths (Figure 2—figure supplement 2 and Figure 3—figure supplement 4): the network topology with high robustness tends to show high dimensionless autocorrelation time when varying regulatory strengths, i.e., accurate oscillation (first ten bars in Figure 2—figure supplement 2 and Figure 3—figure supplement 4); the network topology with low robustness displays a bad performance of oscillation accuracy in the whole parameter space (last ten bars in Figure 2—figure supplement 2 and Figure 3—figure supplement 4). Besides, our work also suggests that tuning network topology is more efficient than changing regulatory strength. This is based on the observations that network topologies with low robustness (last ten bars in Figure 2—figure supplement 2 and Figure 3—figure supplement 4) cannot have a high oscillation accuracy even when searching all kinetic parameter space, but changing topologies may increase the probability of high oscillation accuracy. So we suggest that a feasible way to improve the oscillation accuracy in synthetic networks is to first modify the topology to avoid low-robustness ones and then tune the regulation strength, as illustrated in Figure 5C.”

Reviewer #2 (Recommendations for the authors):The initial analysis was based on oscillation behavior of ODE models at different parameter values. There have been detailed studies in the past with this approach, and the current results appear to be similar, eg. oscillation appear to coincide with the graph cycle structures in the network. The authors may wish to discuss their findings from ODE analysis with previous results of 'Glass, Leon, and Joel S. Pasternack. "Stable oscillations in mathematical models of biological control systems." Journal of Mathematical Biology 6.3 (1978): 207-223.’ and explain the relationship of their findings with these earlier results.

Thank you for the good suggestion. The paper you mentioned modeled the feedback inhibition biological network by piecewise linear equations and derived conditions to generate oscillation. The core motif to produce oscillation is the same as that in our manuscript, and we pointed this out in Page 8 by adding the following text: “Note that these oscillatory network topologies all have a negative feedback structure, which is consistent with previous studies (Glass and Pasternack, 1978; Novák and Tyson, 2008).